

# Five decades of Abramov glacier dynamics reconstructed with multi-sensor optical remote sensing

Enrico Mattea[1], Etienne Berthier[2], Amaury Dehecq[3], Tobias Bolch[4], Atanu Bhattacharya[5,6],
Sajid Ghuffar[7], Martina Barandun[1], and Martin Hoelzle[1]

[1]Department of Geosciences, University of Fribourg, Fribourg, Switzerland
[2]LEGOS, Université de Toulouse, CNES, CNRS, IRD, UT3, Toulouse, France
[3]Univ. Grenoble Alpes, IRD, CNRS, INRAE, Grenoble INP, IGE, 38000 Grenoble, France
[4]Institute of Geodesy, Graz University of Technology, Graz, Austria
[5]Department of Earth Sciences and Remote Sensing, JIS University, Kolkata, India
[6]Centre for Data Science, JIS Institute of Advanced Studies & Research, Kolkata, India
[7]Department of Space Science, Institute of Space Technology, Islamabad, Pakistan

**Correspondence:** Enrico Mattea (enrico.mattea@unifr.ch)

**Abstract.** Reference glacier sites with systematic *in situ* monitoring provide crucial information to understand trends in regional change. Throughout Central Asia, several sites have been established over the past 15 years, often restarting long-term time series interrupted after the Soviet Union collapse. The region also features widespread ice flow instabilities, including surge-type glaciers. Unstable ice dynamics have been usually observed within large-scale remote sensing studies, with limited

ground validation or historical observations. This hampers interpretation of the driving factors of glacier change, their interaction with mass balance, and regional representativity of single glaciers. Here, we reconstruct ice dynamics at the reference Abramov glacier using satellite-based optical remote sensing. The glacier, monitored *in situ* over 1967–1999 and again since 2011, experienced a well-documented episode of fast flow in 1972–1973. We compile a 55-year dataset of digital elevation models (DEMs) and orthoimages by processing raw and analysis-ready imagery from multiple archives. Our estimates for

glacier length and volume changes agree well with previous *in situ*, remote sensing, and model studies. We describe a second unobserved pulsation (2000–2005) at subseasonal scale, not resolved by Landsat or ASTER products. We also measure the buildup to a third active phase, with doubling of mean annual velocity since 2011 despite a continued mass loss of $-0.55 \pm 0.06$ m w.e. yr$^{-1}$. The collected evidence indicates that Abramov is a surge-type glacier with a recurrence time of 20–30 years, challenging its representativity for regional mass balance. However, our results also suggest a potential ongoing transition towards

more stable dynamics.

## 1 Introduction

Glaciers are an essential component of the hydrological cycle in Central Asia, providing a large fraction of the runoff during the dry summer months – a critical contribution for diverse sectors such as agriculture or hydropower production (Huss and Hock, 2018; Hoelzle et al., 2019). At the same time, the vast glacierization across Tien Shan and Pamir, in High Mountain

Asia, presents a multitude of glacier-related hazards such as collapses and glacial lake outburst floods (GLOFs), which can



significantly damage settlements and livelihoods (Komatsu and Watanabe, 2014; Muccione and Fiddes, 2019). Accelerated climate change is impacting glaciers in Central Asia, leading to significant changes in runoff patterns and hazard potential, with consequences still largely uncertain (Unger-Shayesteh et al., 2013; Xenarios et al., 2019; Barandun et al., 2020). Therefore, both long-term monitoring and improved process understanding are crucial to reduce the current uncertainties (Barandun et al.,

25  2020).

The Soviet Union maintained an extensive monitoring program on reference glacier sites in Central Asia, resulting in a comprehensive observational dataset on mass balance, meteorology, hydrology, and ice dynamics. Following the fall of the Soviet Union, this monitoring was mostly discontinued during the 1990s (Unger-Shayesteh et al., 2013); however annual monitoring has been progressively reestablished in recent years at several sites, providing valuable continuation to long-term

measured time series, especially of meteorological variables and glacier mass balance (Schöne et al., 2013; Hoelzle et al., 2017; Barandun et al., 2018). The observational gap in the mass balance records for individual reference glaciers has been addressed with several methods, including geodetic estimation (Denzinger et al., 2021), parameterized approaches such as temperature-index models (Barandun et al., 2015; Kronenberg et al., 2016; Kenzhebaev et al., 2017; Barandun et al., 2018; Popovnin et al., 2021; Van Tricht et al., 2021; Azisov et al., 2022), and full energy-balance simulations (Kronenberg et al., 2022). In recent

years, several regional- and global-scale studies have also estimated mass balance for most or all glaciers in Central Asia, using satellite-based remote sensing to derive surface elevation changes and/or constrain large-scale modeling (Gardner et al., 2013; Gardelle et al., 2013; Farinotti et al., 2015; Kääb et al., 2015; Brun et al., 2017; Zemp et al., 2019; Shean et al., 2020; Hugonnet et al., 2021; Barandun et al., 2021; Fan et al., 2023). Yet, Barandun and Pohl (2023) found data inconsistencies and regional simplifications to hinder interpretation and attribution of the trends shown in such large-scale datasets, and the time series from

reference sites remain crucial for a better understanding of the changing Central Asia cryosphere.

Mass balance representativity of reference glaciers can be affected by their ice dynamics (Oerlemans and Van Pelt, 2015; Lv et al., 2020b). The mountains of Central Asia present a high prevalence of glaciers with unstable ice flow (Sevestre and Benn, 2015; Mukherjee et al., 2017; Goerlich et al., 2020; Guillet et al., 2022), known as pulsating glaciers in Soviet literature and as surge-type glaciers in Western terminology (Jiskoot, 2011). Such glaciers periodically alternate between active phases

of accelerated flow, with downstream mass redistribution, and longer stretches of quiescent recovery. The cycle is driven by an intrinsic dynamic instability (Meier and Post, 1969; Sevestre and Benn, 2015), although climatic forcing and mass balance can partly shape the characteristics and duration of the active phase (Emelyianov et al., 1974; Eisen et al., 2001; Hewitt, 2007; Flowers et al., 2011; Pitte et al., 2016). Several mechanisms have been proposed to explain the occurrence of glacier surges (Kamb, 1987; Fowler et al., 2001; Harrison and Post, 2003; Benn et al., 2019; Thøgersen et al., 2019; Terleth et al., 2021);

yet, unstable ice dynamics in Central Asia appear to elude classification according to the classical models (Quincey et al., 2015; Chudley and Willis, 2019; Lv et al., 2020a). Moreover, differentiation between a surge and an advance is not always straightforward (Paul, 2015; Lv et al., 2020a), as a continuous spectrum of flow instabilities is assumed to exist between the stable and surge-type end-members (Mayo, 1978; Jiskoot, 2011; Herreid and Truffer, 2016). Unstable glaciers present challenges for the monitoring and interpretation of mass balance trends (Bhambri et al., 2017). Glacier-wide mass balance can

fluctuate from a positive anomaly before the active phase to a negative one immediately after, as the glacier abruptly advances





to lower and warmer locations (Lv et al., 2020b; King et al., 2023). The altitudinal mass balance gradient tends to be too steep during quiescence, compensated by downward mass redistribution during the active phase (Dolgoushin and Osipova, 1975; Raymond, 1987; Lv et al., 2020b). Increased water storage within and beneath the glacier can significantly affect geodetic mass balance estimations (Raymond, 1987; Humphrey and Raymond, 1994). Rapid evolution of the surface – for example,

area, slope, and albedo affecting the energy balance – is not easily accounted for within model studies (Emelyianov et al., 1974; Kronenberg et al., 2022). Finally, the monitoring network can be damaged or become inaccessible during the active phase, which can be especially problematic for sites with complex logistics or infrequent revisits (Emelyianov et al., 1974). Therefore, mass balance trends of surge-type glaciers are generally recognized as representative only over a timescale longer than one surge cycle, and these glaciers are usually excluded from (or treated separately within) geodetic estimates of regional

mass balance (Gardelle et al., 2013; Zhou et al., 2019; Lv et al., 2020b).

A key reference site in Central Asia potentially affected by unstable ice dynamics is Abramov (39°37' N, 71°33' E; Fig. 1), a valley glacier in the Pamir-Alay mountains of Kyrgyzstan covering 21 km$^2$ as of 2024. Between 1967 and 1999, the Abramov glacier was monitored year-round at a dense network of point measurements, resulting in a detailed dataset of meteorological, glaciological, and hydrological observations (Suslov et al., 1980; Glazirin' et al., 1993; Pertziger, 1996). Glacier-wide calcu-

lations by various authors have indicated a mean mass loss of 0.4 to 0.7 m w.e. yr$^{-1}$ over the monitored period, with almost no year of mass gain (Barandun et al., 2015). Between 1972 and 1973, after several years of acceleration, the glacier experienced a pulsation: surface velocity increased up to five-fold and the terminus advanced more than 400 m, with a maximum velocity of 2 m d$^{-1}$ (Emelyianov et al., 1974; Kotlyakov et al., 2017). Flow velocity on the glacier tongue peaked in June 1973, before entering a rapid decline down to a minimum reached in early 1974 (Abulkhasanova et al., 1979; Suslov et al., 1980).

This low-intensity event was characterized as a quiet pulsation, with relatively minor mass redistribution along the glacier, as opposed to stronger traditional surges (Glazirin', 1978; Mayo, 1978; Glazirin' and Shchetinnikov, 1980). Soviet and Russian publications have reported somewhat contrasting measurements for this event, including terminus retreat in the preceding years, total advance during the active phase, ice volume transferred to the tongue, and estimated recurrence time (Emelyianov et al., 1974; Suslov et al., 1980; Dolgoushin and Osipova, 1982; Glazirin' et al., 1993; Kotlyakov et al., 2017); still, all agree

on the general characteristics. Using a minimal flow-line model, Glazirin' et al. (1987) qualitatively interpreted the pulsation as the result of a sudden shift in basal conditions. The subsequent recovery period lasted into the 1990s, with slow and regular ice flow and widespread thickness loss over the glacier tongue (Abulkhasanova et al., 1979; Suslov et al., 1980; Grishin and Abulkhasanova, 1986; Glazirin' et al., 1993). The Abramov research station was abandoned in 1999 (Savoskul, 2000), and the glacier went unobserved until 2011, when annual mass balance monitoring was resumed and an automatic weather station

(AWS) was installed (Schöne et al., 2013; Hoelzle et al., 2017). Unlike for mass balance, the gap in ice dynamics measurements has not been addressed so far, and systematic monitoring has not been reestablished.

Here we reconstruct 55 years of unstable ice flow at Abramov with satellite-based optical remote sensing. We focus on the years of the observed pulsation (early 1970s), the period of interrupted *in situ* monitoring (2000s), and the present-day evolution. We find that existing satellite-based datasets such as Landsat and ASTER fail for the most part to resolve ice

dynamics at Abramov, and thus we process several scenes from multiple higher-resolution sensors, such as the Indian Remote



Sensing (IRS-1C/D) program (Kasturirangan et al., 1996), the Satellite Pour l'Observation de la Terre (SPOT; Riazanoff, 2002) and the RapidEye constellation (Tyc et al., 2005). We measure changes in glacier thickness, velocity, and terminus position, describing a second unobserved pulsation in the early 2000s and the present-day buildup to a third one.

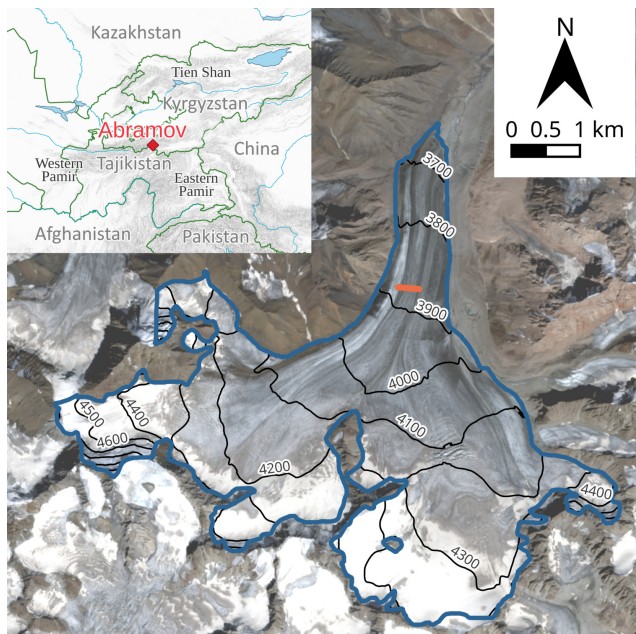

**Figure 1.** Location and detail maps of Abramov glacier. Background: Sentinel-2 multispectral instrument (MSI) natural color from 5 September 2022. Elevation contours: Pléiades digital elevation model (DEM) from 5 September 2022. The orange line indicates transverse profile 4 of the historical monitoring network.

## 2 Materials and Methods

### 2.1 Analysis-ready data


Several orthoimages and topographic datasets are available for Abramov glacier from public archives and previous studies (Table 1). We downloaded all Level-1C orthoimages available from the Sentinel-2 repository (Drusch et al., 2012) between 2015 and 2023, specifically bands 2, 3, 4, and 8 at 10 m resolution. The entire glacier surface is covered by two relative orbits (91 and 134), providing four revisits every 10 days since 2017. We used these scenes to derive surface ice velocities at annual fre-

quency. We also collected the data from Denzinger et al. (2021) – orthoimages and DEMs of the glacier obtained from scanned aerial frames from 1975 – and a submeter stereo pair captured by Pléiades in 2015 (Berthier et al., 2014). From Hugonnet et al. (2021), we obtained a set of DEMs generated at 30 m resolution out of all ASTER stereo pairs (NASA/METI/AIST/Japan Spacesystems and U.S./Japan ASTER Science Team, 2001); four such DEMs are cloud- and snow-free at Abramov and cover the early 2000s. In addition, we retrieved elevation grids from two radar-derived global DEMs: Copernicus (European Space



Agency and Airbus, 2022) and NASADEM (NASA JPL, 2020). The Copernicus DEM is downsampled from higher-resolution data and shows better detail and lower error metrics than those from NASADEM when compared with reference measurements (Fahrland, 2022; Li et al., 2022; Okolie et al., 2024). However, a large-scale elevation artifact in the vicinity of Abramov glacier prevents direct use of this DEM within comparisons of surface change, and thus we chose to use it only (in the GLO-30 and GLO-90 versions) as initial seed for stereo DEM generation. Conversely, NASADEM provides surface elevation data from February 2000 with somewhat higher noise but no apparent artifacts, and thus we chose to use it (1) to compute local changes of ice thickness, (2) as coregistration reference for all DEMs, and (3) to provide the topography for orthorectification of monoscopic scenes.

In addition to surface DEMs, bed elevation for the Abramov glacier is also known: Kuz'michenok et al. (1992) compiled a topographic map at a scale of 1:25000, depicting the bed with contour lines at 10 m intervals. The original data were computed by combining the surface topography (measured by aerial survey) with a dense array of ice thickness measurements collected by radio-echo sounding in 1986. We georeferenced this map using 20 ground control points (GCPs) manually extracted from the 2015 Pléiades DEM, then digitized all contour lines and interpolated them into a bed DEM at 30 m resolution, using regularized spline with tension (Mitášová and Hofierka, 1993).

## 2.2 Newly produced datasets

We found that existing datasets do not provide an adequate level of detail to accurately reconstruct the dynamics of a relatively small and slow-moving glacier such as Abramov, especially for the period of interrupted monitoring (Appendix E). Therefore, we processed additional archived satellite scenes from multiple sensors to improve the spatio-temporal coverage of both orthoimages and DEMs.

We collected 35 monoscopic scenes from SPOT satellites 1 through 5, acquired between 1986 and 2013 and openly distributed since 2020 within the SPOT World Heritage (SWH) program (Nosavan et al., 2020). The scenes have radiometric resolution of 8 bit and nominal spatial resolution at nadir of 20 m (SPOT 1–4 in multispectral mode), 10 m (SPOT 1–4 in panchromatic mode; SPOT 5 multispectral), and 5 m (SPOT 5 panchromatic). We obtained 14 panchromatic scenes by IRS-1C/D through the Antrix Corporation Limited (Antrix Corporation Limited, 2019) recorded at 5.7 m pixel size and 6 bit depth between 1997 and 2006. Finally, we obtained 11 raw multispectral scenes from the RapidEye constellation (Tyc et al., 2005) through the Education and Research Program of Planet Labs (Planet Labs, 2024). The scenes feature 6.5 m resolution at 12 bits and cover the period 2010–2019. Processing of all monoscopic scenes is described in Sect. 2.2.1.

For DEM generation (Sect. 2.2.2), we processed 18 stereo pairs from the SPOT 5 High-Resolution Stereoscopic (HRS) instrument, also distributed through the SWH program, covering the period from 2003 to 2006. These images feature rectangular pixels with a nominal ground sampling distance of 5 by 10 m. For the period of the 1970s pulsation, we produced DEMs and orthophotos from five stereo and two tristereo KH-4B and KH-9 panoramic camera (PC) and mapping camera (MC) scenes, acquired between 1968 and 1980 within the US spy satellite programs Corona and Hexagon (Table A2). The declassified film scans are available from the US Geological Survey (Earth Resources Observation And Science (EROS) Center, 2017a, b, c).



**Table 1.** Analysis-ready data from previous studies and public archives. HiRI: high-resolution imager. MSI: multispectral instrument. PAN: single-band panchromatic.

| Data type | Platform and sensor | Resolution | Time coverage | Notes | References |
|---|---|---|---|---|---|
| Orthophoto | Aerial, AFA-TES camera | 2 m | 12/07/1975 | | Denzinger et al. (2021) |
| | Pléiades HiRI PAN | 0.5 m | 01/09/2015 | | Barandun et al. (2018) |
| | Sentinel-2 MSI | 10 m | 2015–2023 | Bands 2/3/4/8 | Drusch et al. (2012) |
| DEM | Aerial, AFA-TES camera | 4 m | 12/07/1975 | | Denzinger et al. (2021) |
| | Terra ASTER | 30 m | 2001–2006 | | Hugonnet et al. (2021) |
| | Space Shuttle, radar (C-band) | 1" | 02/2000 | NASADEM | NASA JPL (2020) |
| | Pléiades HiRI | 4 m | 01/09/2015 | | Barandun et al. (2018) |
| | TanDEM-X, radar (X-band) | 1", 3" | 2011–2015 | Copernicus GLO-30 and GLO-90 | European Space Agency and Airbus (2022) |

**Table 2.** Newly processed satellite datasets. GSD: best-case ground sampling distance of raw scenes. PC: panoramic camera. MC: mapping camera (tristereo). HRS: high-resolution stereoscopic. HRV: high-resolution visible. HRVIR: high-resolution visible and infraRed. HRG: high-resolution geometric. REIS: RapidEye Earth-imaging System. PAN: single-band panchromatic. MS: multispectral.

| Data type | Platform | Sensor | GSD [m] | Time coverage | Count |
|---|---|---|---|---|---|
| Stereo pairs | Corona | KH-4 PC | 1.8 | 18/08/1968 | 1 |
| | Hexagon | KH-9 MC | 6 | 1973–1980 | 3 |
| | Hexagon | KH-9 PC | 0.6 | 1972–1980 | 3 |
| | SPOT 5 | HRS | 5x10 | 2003–2011 | 18 |
| | Pléiades | HiRI PAN | 0.5 | 2020–2023 | 4 |
| Mono scenes | SPOT 1 | HRV MS | 20 | 1986 | 1 |
| | SPOT 3 | HRV PAN | 10 | 1996 | 1 |
| | SPOT 4 | HRVIR PAN/MS | 10/20 | 1999–2003 | 8 |
| | SPOT 5 | HRG PAN/MS | 5/10 | 2003–2013 | 18 |
| | IRS-1C/D | PAN | 5.7 | 1997–2006 | 14 |
| | RapidEye | REIS | 6.5 | 2010–2019 | 11 |

Finally, we processed four additional Pléiades stereo pairs at 50 cm resolution acquired between 2020 and 2023. All newly processed datasets are presented in Table 2.



### 2.2.1 Orthorectification of monoscopic scenes

We produced orthoimages from SPOT, IRS, and RapidEye with different tools, depending on the supporting metadata available within each dataset. We used the 50 cm Pléiades scene from September 2020 as reference for horizontal alignment, owing to its minimal cloud and snow cover. This scene extends over the region of Abramov glacier with a footprint of approximately 18 by 20 km.

The monoscopic SPOT scenes were orthorectified over the NASADEM terrain using the SPOT-specific rigorous sensor model (RSM) of Aati et al. (2022), as implemented in the open-source geoCosiCorr3D package. This method involves refinement of image geometry using GCPs to correct errors associated with the satellite viewing parameters reported in the scene metadata (Leprince et al., 2007; Aati et al., 2022). Technical details on SPOT orthorectification, including our adaptations of the pipeline of Aati et al. (2022), are described in Appendix B.

We received scenes from IRS-1C/D in raw format, with basic geometric corrections but no metadata or information on satellite viewing parameters. We orthorectified them using the sensor model of Toutin (2004), as implemented in Catalyst Professional Orthoengine (PCI Geomatics Enterprises, 2024). After providing a coarse geolocation from five manual GCPs, we used a 2006 SPOT 5 HRG orthophoto as reference for automatic GCP generation. We applied the Smallest Univalue Segment Assimilating Nucleus (SUSAN) algorithm (Smith and Brady, 1997) for feature point detection, then fast Fourier transform phase matching (Kuglin and Hines, 1975) for tie point calculation. Subsequent orthorectification is basically independent of the manual choice of initial GCPs. We produced final orthoimages at 5 m resolution.

Finally, RapidEye scenes were orthorectified by means of the mapproject tool from the NASA Ames Stereo Pipeline (ASP; Beyer et al., 2018), based on the rational polynomial coefficient (RPC) information provided within each product. We used the image_align tool from the same software package to apply fine registration of each orthophoto to the SPOT 5 HRG reference, with a rigid transform estimated from interest point matching with singular value decomposition.

### 2.2.2 Stereo processing

We processed stereo scenes into DEMs and orthoimages, in most cases, using the open-source tools from ASP, with specific processing steps and parameters adapted to the characteristics of each stereo dataset. The only exceptions were the three scenes acquired by panoramic cameras onboard the KH-4B and KH-9 satellites (Table 2). Due to the complex acquisition geometry, we processed the KH-4B data with the Remote Sensing Software Package Graz (RSG) using the methods in Bhattacharya et al. (2021) and the KH-9 PC scenes according to Ghuffar et al. (2022, 2023). Spatial resolution of the original images is 1.8 and 0.6 m for KH-4B and KH-9, respectively. We also processed imagery from the mapping camera onboard KH-9, with the automated method in Dehecq et al. (2020), again based on NASA ASP. KH-9 PC and MC acquisition dates do not match (Table 2) because the two systems were not usually operated at the same time.

We produced 15 DEMs from the SPOT 5 HRS stereo pairs at 30 m resolution. We ran stereo correlation with the More Global Matching (MGM) algorithm (Facciolo et al., 2015) as implemented in NASA ASP and the processing parameters of Shugar et al. (2021) and Bhushan and Shean (2021). In four scenes the glacier is split over two pairs, which we mosaicked into single



DEMs. Our dataset also includes four instances of repeat HRS acquisitions at an interval of just 5 days. The corresponding DEM pairs were used to verify the stability of our results on glacier terrain by measuring the mean change in surface height

over the outlines of the Randolph Glacier Inventory (RGI) version 7 (RGI 7.0 Consortium, 2023). Further methodological details on SPOT DEM generation are described in Appendix C.

Finally, we generated Pléiades DEMs at 4 m resolution for September 2020, 2022, and 2023. We used the semi-global matching (SGM) algorithm with parameters from Deschamps-Berger et al. (2020) both for consistency with the 2015 Pléiades DEMs of Barandun et al. (2018) and for a lower computational cost compared to that of MGM (each scene includes more than

$10^9$ pixels). We then orthorectified the raw panchromatic images over their respective DEM surfaces using the provided RPC metadata to a final pixel size of 0.5 m. The 2022 dataset consists of two half-pairs split longitudinally, which we mosaicked with the same methods as for HRS (Appendix C).

### 2.3  Derivation of glaciological parameters

#### 2.3.1  Glacier terminus mapping

We manually digitized the terminus of Abramov glacier on all cloud-free orthoimages in our dataset; availability of a single band (in most cases) prevented the application of automated ice mapping methods, which would require, in any case, manual corrections to obtain the highest accuracy (Racoviteanu et al., 2009; Paul et al., 2015). We used the digitized outlines as aggregation polygons to compute changes of ice volume and applied the rectilinear box method to compute evolution of the width-averaged glacier length (Moon and Joughin, 2008; Lea et al., 2014). To quantify length uncertainty from residual

misregistration and distortions, the method was repeated on two ostensibly stable features: a rock ridge and an unpaved road (Fig. 2b). The two features have similar length to the width of the glacier terminus, are located within 1 km of it, and, being almost perpendicular, account for misregistration in both the latitudinal and longitudinal directions. Uncertainty of the glacier length was then expressed as the maximum deviation among the two features from the mean position of each of them computed across all the orthoimages.

#### 2.3.2  Surface ice velocity


We derived annual surface ice velocities with two different approaches based on image correlation: for 1996–2020, we processed single pairs of summer orthoimages (generated in Sect. 2.2); for 2017–2023, we produced annual mosaic combinations based on the dense Sentinel-2 archive.

The single summer orthoimages were manually selected for minimum snow and cloud cover on the glacier; the scenes are

available for all years from 1996 to 2020, with gaps in 1999 and 2008–2009 (Table A1). We computed east/west and north/south displacements using the frequency correlator of geoCosiCorr3D, with a 128 px window size (Leprince et al., 2007; Aati et al., 2022). To increase coverage, we also correlated scenes from different sensors (SPOT with IRS; RapidEye with Pléiades), bilinearly resampling to 5 m where needed. To improve correlation quality, we preprocessed the images with contrast limited adaptive histogram equalization (CLAHE; Pizer et al., 1987; Van Wyk De Vries and Wickert, 2021). We then applied common



filters to the displacement maps (e.g., Heid and Kääb, 2012; Millan et al., 2019; Beaud et al., 2022): we discarded pixels of low signal-to-noise ratio (below 0.97), excessive velocity magnitude (above 200 m yr$^{-1}$; Fig. 4a), and small isolated specks (fewer than 100 contiguous cells). We also detected outliers with an $11 \times 11$ moving-window median filter, thresholded at two standard deviations, and subtracted the median east/west and north/south displacements over stable terrain to account for residual misregistration. We finally extracted along-flow velocity profiles on the glacier tongue (Fig. 4b) and filtered out

longitudinal strain rates above 0.1 yr$^{-1}$ (e.g., Zheng et al., 2023), based on the values reported by Emelyianov et al. (1974) for the fast flow episode of 1972–1973. Velocity derivation from SPOT and IRS was possible only on the lower tongue due to sensor saturation on snow-covered terrain. For each correlated pair, uncertainty was estimated as the normalized mean absolute difference (NMAD) of velocity measured over stable terrain (Scherler et al., 2008; Millan et al., 2019; Paul et al., 2022). The image pairs cover slightly inconsistent time intervals, with acquisitions ranging from 26 July to 15 September; a seasonal cycle

in surface velocity could then introduce a bias in our annual results. This effect was accounted for by conservatively assuming velocity deviations by $\pm\,50\,\%$ of the annual average over a number of days equal (for each pair) to the difference between 365 and the actual interval. The resulting uncertainty range was then quadratically summed with the NMAD (Fig. 4a).

    Over 2017–2023, we derived surface flow velocities from the Sentinel-2 archive. Based on the cloud masks provided with each Sentinel-2 product, we discarded all acquisitions showing clouds on more than 10 % of the glacier surface. We prepro-

cessed the scenes with CLAHE followed by an orientation filter (Fitch et al., 2002; Van Wyk De Vries and Wickert, 2021); the latter further improved correlation between Sentinel-2 scenes, unlike for the 6- and 8-bit data from SPOT and IRS. We then ran the image pairs through the frequency correlator of geoCosiCorr3D separately within each of the two orbital tracks and four spectral bands (Sect. 2.1). For correlation, we included all pairs with time separations of 5 to 100 and 300 to 430 days (Millan et al., 2019). We obtained approximately 68 000 displacement maps with a resolution of 40 m. The maps were filtered

with signal-to-noise ratio and magnitude thresholds as above, but (for computational performance) we omitted the other filters applied to single image pairs. Instead, after subtraction of stable-terrain residual displacements, the east/west and north/south velocity maps were stacked within each year, computing velocities for each pixel as the median of all available values. Results of the four bands were finally combined using arithmetic average, weighted by the number of scene pairs contributing to each pixel. The final results were two sets of annual velocity maps – one per relative orbit – representative of surface velocity

from one summer to the next. Note that Sentinel-2 products are disseminated with different processing baselines depending on acquisition date (European Space Agency, 2024). A noteworthy processing inconsistency is the use of different DEMs for orthorectification (Kääb et al., 2016): initially Planet DEM90, replaced by Copernicus GLO-90 in 2021, and Copernicus GLO-30 in 2023. Other relevant changes include the activation of geometric refinement using a global reference image for GCPs (since 2021) and a recalibration of the Sentinel-2B sensor radiometry to that of Sentinel-2A since 2022. As of early 2024, the

full archive is undergoing backwards reprocessing to a common baseline, featuring all the improvements introduced over the years; thus, the more recent scenes are available in two versions. We verified the minor impact of such processing changes by running our pipeline twice: over the original baseline and over the reprocessed one. Considering all Abramov glacier cells faster than 10 m yr$^{-1}$ (cf. Fig. 4c/d), the resulting absolute differences in surface velocity magnitude were within 5 m yr$^{-1}$ in more than 99 % of all pixels. Hereafter, we retain the results obtained with reprocessed scenes whenever available.



### 2.3.3  Surface elevation and volume changes


We computed surface elevation changes from DEM differences. We implemented a uniform pipeline for the processing and analysis of all DEM pairs of interest, with time separations ranging from 5 days to more than 30 years. All grids were initially cropped to the Abramov glacier region and bilinearly resampled to a common projection (Universal Transverse Mercator zone 42N) and resolution (30 m, except for comparisons of the Pléiades high-resolution DEM pairs which were kept at the original

4 m). Then, the more recent DEM of the pair was coregistered to the older one using the approach in Nuth and Kääb (2011), followed by a 2D tilt correction (Girod et al., 2017) and the subtraction of residual median vertical offset over stable terrain. Such terrain was delimited by exclusion of (1) the RGI outlines, buffered by 200 m; (2) the slopes above 30° (Denzinger et al., 2021); and (3) all outlier cells whose absolute difference before coregistration was greater than four NMADs (Hugonnet et al., 2022). After coregistration, the DEM difference was first filtered (within the glacier outline) using simple thresholds on the

absolute change. Based on long-term *in situ* observations (Suslov et al., 1980), we used conservative threshold values of 200 and 50 m for the lower tongue and upper region, respectively, separated at 4000 m asl (Fig. 1). Since the largest elevation changes occur far away from such a demarcation line (Fig. 5a, b, e, f), sensitivity to its chosen altitude is very low. As a second step, we quantified across- and along-track elevation biases (Hugonnet et al., 2022; Falaschi et al., 2023). We computed the along-track coordinate of each grid cell as follows:

$$x_{al} = -x \cdot sin(\theta) + y \cdot cos(\theta) \tag{1}$$

where $x$ and $y$ are the column and row numbers, respectively, of the cell and $\theta$ is the ground track of the satellite; across-track coordinate is simply perpendicular and was expressed similarly. We then computed elevation differences on stable terrain and calculated the mean difference within bins of the across/along-track coordinate. We corrected some common biases using polynomial or spline fits. Specifically, we compensated the track biases of Pléiades as in Falaschi et al. (2023); we also mitigated

KH-9 MC DEM artifacts from film scanning (Dehecq et al., 2020) and the residual artifacts of KH-4B and KH-9 PC using, in both cases, along-track spline fits. Following Nuth and Kääb (2011) and Gardelle et al. (2012), we also quantified (using the same method) biases related to maximum and minimum terrain curvature, absolute altitude, slope, and aspect; we simply verified that the mean magnitude of such biases did not exceed the spread observed for each bin, but no correction was applied to the grids. After a second coregistration of all the bias-corrected DEMs, we filtered the DEM differences using a threshold of

three standard deviations within 100 m elevation bands (Berthier et al., 2004; Ghuffar et al., 2023). Then, the mean gap-filled elevation change was calculated using the local hypsometric method (McNabb et al., 2019). We considered regions of interest both the entire glacier and some smaller polygons covering the tongue and terminus areas (Fig. 5). Finally, uncertainties in the elevation changes were estimated using the stable-terrain analysis in Hugonnet et al. (2022). We first calculated the dependence of DEM error on slope and maximum curvature and used this to compute standardized uncertainties and compile a

map of pixel-wise elevation error for each DEM difference. Then, we modeled the variogram of elevation error with a double-range exponential function to compute the number of effective samples of elevation difference and finally the uncertainty of integrated elevation change (Hugonnet et al., 2022). Uncertainty estimates were validated using Monte Carlo spatial sampling



with circular patches (Berthier et al., 2016; Miles et al., 2018; Hugonnet et al., 2022). In a few cases, we observed a slight under-estimation of uncertainty from variogram modeling compared with the patches method. We accounted for such a discrepancy

by fitting a small fraction of fully correlated variance (average: 1.0 %) to the variogram result, until the two estimates matched (Mannerfelt et al., 2024). Following Berthier et al. (2014) and Dussaillant et al. (2018), a conservative factor of 5 was used to multiply uncertainty of elevation change over data gaps.

While our study is focused on remote sensing of ice dynamics, several of our DEMs present almost complete spatial coverage of the glacier, including the accumulation area. We used them to provide geodetic mass balance estimates for six subperiods

between 1973 and 2022, assuming a mean glacier density of 850 kg m$^{-3}$ (Huss, 2013). Unlike for local ice thickness changes, for glacier-wide mass balance we only considered optical DEMs to avoid biases from radar penetration in the upper firn area (Barandun et al., 2015; Dehecq et al., 2016; Fan et al., 2023). Acquisition dates of the selected DEMs range from 12 July to 29 November, rendering necessary to perform seasonality correction of geodetic volume changes (Cox and March, 2004; Cogley, 2009). This correction was computed by adding or subtracting the respective glacier-wide mass balance derived from

the calibrated daily simulation in Kronenberg et al. (2022); correction values range from $-0.03$ to $+0.24$ m w.e. yr$^{-1}$. We computed mass balance uncertainty following Dussaillant et al. (2018), using standard error propagation of the uncertainties of (1) integrated elevation change, including data gaps (Dussaillant et al., 2018; Hugonnet et al., 2022); (2) glacier area (5 %; Paul et al., 2013); (3) density conversion (60 kg m$^{-3}$; Huss, 2013); and (4) seasonal correction (0.7 m w.e. yr$^{-1}$; RMSE between modeled and measured mass balance in Kronenberg et al., 2022, which we linearly rescaled from 1 year to the actual number

of days of the seasonal correction for each subperiod). Uncertainty of density conversion over short periods is further discussed in Sect. 4.3.

We checked consistency of our remote sensing data over three annual intervals, between 2003 and 2006, at the glacier terminus region by combining simultaneous measurements of volume changes with mean surface velocity, absolute ice thickness, and modeled surface mass balance. This calculation is presented in Appendix D.

## 295 3  Results

### 3.1  Glacier length changes

We obtained a set of 40 terminus positions and width-averaged length changes over 1968–2023 (Fig. 2). The time separation between consecutive measurements (Fig. 2a) ranges from 21 days (during the early 2000s) to 10 years (1986 to 1996). The terminus shows an overall trend of retreat over the 55 years, with a total change of $-1106 \pm 4$ m, at a mean rate of $20.11 \pm 0.07$

m yr$^{-1}$. This trend was interrupted by two sudden transitions to rapid front advance, in 1972 and 2002. The second advance was not observed *in situ* as this occurred just after the permanent monitoring program was discontinued. For both advances, the transition corresponded to the arrival of a wave of active ice, overrunning the remnants of a stagnant debris-covered tongue. Total advance during the two events was $350 \pm 6$ m from August 1972 to November 1973, and $151 \pm 2$ m from August 2002 to January 2006. However, observation of the exact dates of minimum and maximum length could be missing from our dataset.

We derived maximum advance rates of $82.3 \pm 0.3$ cm d$^{-1}$ in 1972–1973 and $44.3 \pm 0.8$ cm d$^{-1}$ in summer 2003. Seasonal



oscillations are visible during the more recent event, with an alternating pattern of marked advance during the cold season and smaller retreat in summer. Both advances were followed by an initially slow but gradually accelerating front retreat. Length uncertainties from the digitized stable features are always within ± 10 m, with standard deviations of 1.7 and 3.2 m for the unpaved road and the ridge, respectively (Fig. 2c).

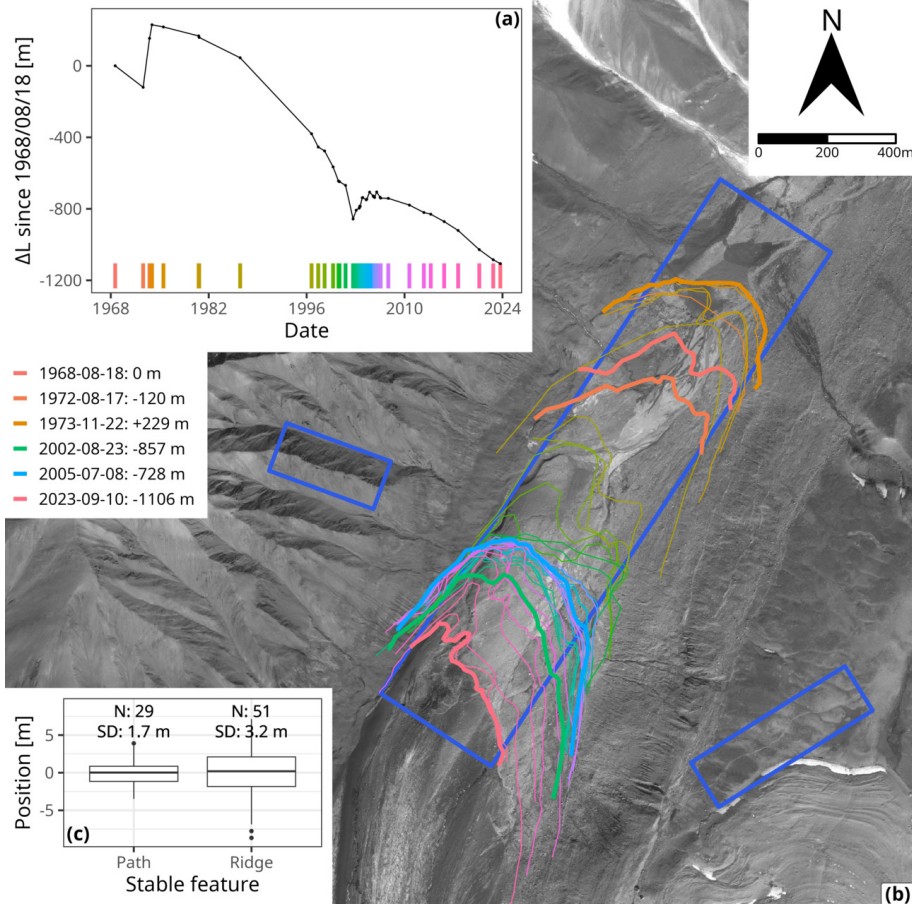

**Figure 2. (a)** Length changes of Abramov glacier according to the rectilinear box method. Zero reference is 18/08/1968. Marks at the bottom match date and color of the outlines in panel **(b)**. The six listed dates (with their cumulative length changes) correspond to the extreme positions of the two pulsations and the earliest and latest points in our dataset; the respective outlines are shown in bold in panel **(b)**. **(b)** Digitized termini of Abramov glacier from 1968 to 2023. Blue rectangles correspond to the rectilinear boxes used to calculate length changes of the glacier and the stable features. Background: Pléiades © CNES 2023, Distribution AIRBUS DS. **(c)** Box-plots of relative deviations of the stable feature positions from their mean.





## 3.2 Surface velocity

Glacier velocities are shown in Fig. 3 for transverse profile 4 of the historical monitoring network (Fig. 1). The profile is located about 2.2 km upstream of the 2023 terminus and presents the most complete results in our dataset. The time series shows a marked acceleration occurring just before and in the early stages of the 2000s advance: mean annual velocity increased regularly between 1996/97 and 2002/03, almost doubling from 57 to 106 m yr$^{-1}$. Afterwards, a gradual (but faster) slowdown is visible, continuing at least into 2006/07. Minimum velocity was probably reached between 2007 and 2010; since then, a new slow regular acceleration has occurred, leading again to a doubling in velocity by 2023 (from 17 to 37 m yr$^{-1}$). Velocity changes at other transverse profiles of the historical monitoring network are qualitatively similar, but with more data gaps. The strong multiannual velocity variations did not reflect corresponding changes in ice thickness (Fig. 3): at profile 4, the latter remained constant (within uncertainty) during the early 2000s, and has gradually lost 18 ± 4 m (10 %) since 2011. We found some specific differences between the early 2000s and the previous speed-up event of 1972–1973: our results show a progressive acceleration leading to a moderate velocity peak (108 ± 2 m yr$^{-1}$; about half the 1973 value) sustained between 2000 and 2003 and a gradual slowdown in subsequent years; by contrast, *in situ* observations of the early 1970s recorded a faster speed-up (with a five-fold increase in the 2 years before the peak), a shorter duration of maximum velocities (about 1 year), and an especially abrupt termination, occurring within less than 10 months (Suslov et al., 1980). Minimum velocities also appear to be lower after the first event, although values for 2007–2010 are missing from our dataset.

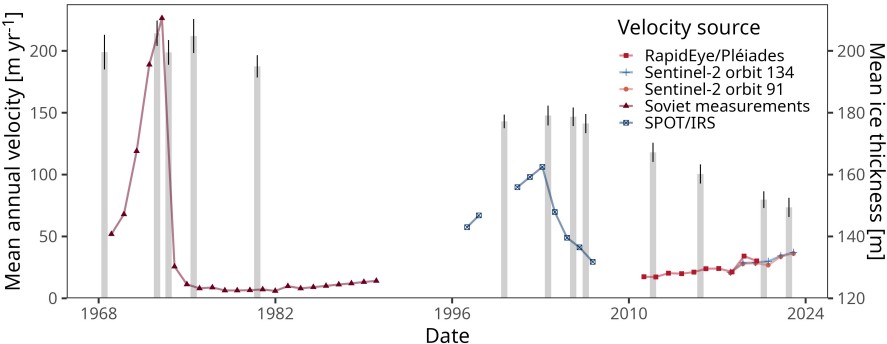

**Figure 3.** Mean annual surface ice velocity at transverse profile 4 (Fig. 1). Soviet measurements are from Glazirin' et al. (1993). Grey bars show DEM-derived mean ice thickness at the same profile. Error intervals reflect only uncertainty from the surface DEMs; uncertainty in bed elevation – which affects all thickness values equally – is not included.

Longitudinal velocity profiles over 1996–2007 (Fig. 4a) show uniform acceleration and slowdown over the entire glacier tongue; only the first profiles are slower towards the terminus, confirming the presence of stagnant ice which, by 2000/01, was being reached by the active front. The more recent velocity profiles (Fig. 4c, d) extend into the right branch and accumulation area thanks to the superior radiometric specification of Sentinel-2 MSI compared with that of older sensors. We observe acceleration since 2017/18 to be again quite uniform along the glacier; however, on the right branch, it is visible only downstream of





the icefall at 4150 m asl (Fig. 1, 4b, d). Flow velocities appear to have peaked by 2022/23, when a slight reduction of 2 m yr$^{-1}$ is visible over the upper left branch compared with the previous year. Conversely, values on the lower tongue still increased by about the same amount over 2023, suggesting a down-glacier propagation of the wave of maximum velocity. In the full map of glacier surface velocity (Fig. 4b), the different flow regions are visible and match the boundaries defined by medial moraines

(Fig. 1). The slower flow of small nearby glaciers is also resolved.

Computed uncertainties of single-pair surface velocities (Fig. 4a) are 4.3 m yr$^{-1}$ on average, but as low as 0.6 m yr$^{-1}$ for 2005–2006. In the annual Sentinel-2 mosaics, NMAD over stable terrain (defined as in Sect. 2.3.3) is always between 0.5 and 1.5 m yr$^{-1}$. Over 2017–2020, the RMSE between Sentinel-2 mosaics and single annual correlations is 3.5 m yr$^{-1}$ at transverse profile 4 (Fig. 3). Plausibility and significance of such estimates is discussed in Sect. 4.3.

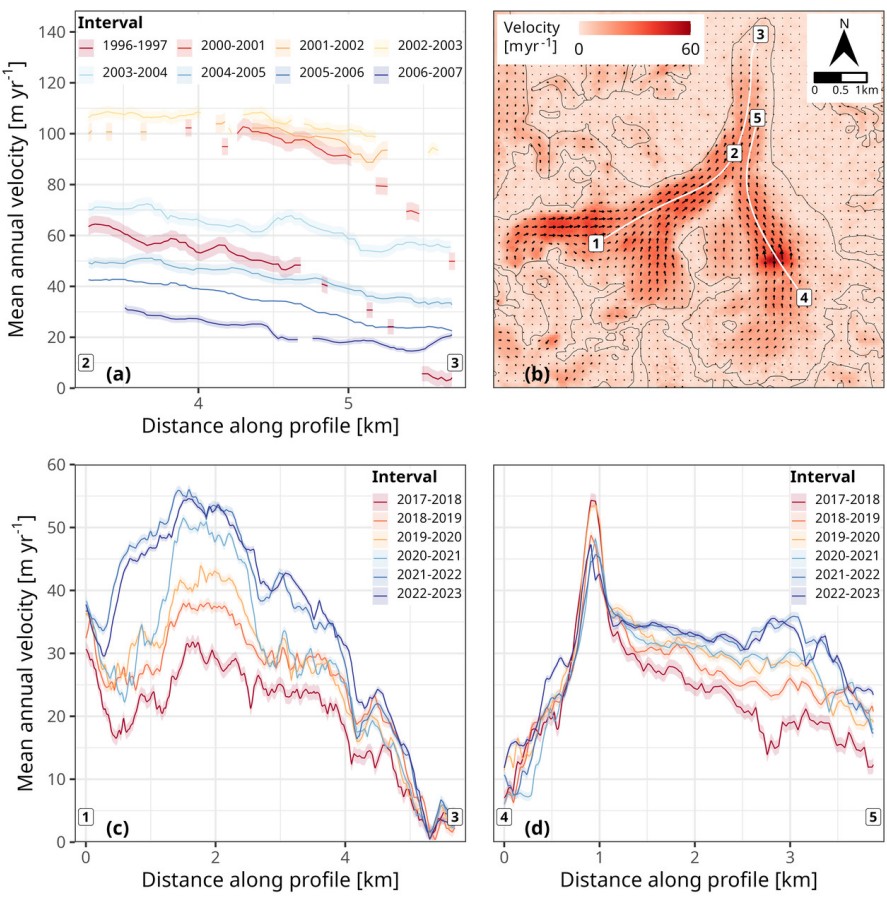

**Figure 4.** Longitudinal profiles of mean annual surface ice velocity from **(a)** SPOT and IRS single pairs, lower tongue; **(c)** Sentinel-2 orbit 91 mosaic, left branch and lower tongue; **(d)** Sentinel-2 orbit 91 mosaic, right branch. **(b)** Mean 2017/18 velocity from Sentinel-2 orbit 91 mosaic; white profiles and locations 1 to 5 correspond to panels **(a)**, **(c)**, and **(d)**. Uncertainty shading reflects the residual velocity of stable terrain (Sect. 4.3).





### 3.3 Surface elevation and volume changes

#### 3.3.1 Thickness patterns

Our DEM differences follow the evolution of the glacier surface over the course of both abovementioned pulsations (Fig. 5). During the active phases (1972–1973 and 2000–2003; Fig. 5a, e), ice thickness significantly increased at the terminus, reaching $90 \pm 5$ m in the first phase and $39 \pm 4$ m in the second one. Within the same periods, a modest albeit widespread thinning is visible over an upstream reservoir region, reaching $20 \pm 5$ in the 1970s event and $10 \pm 3$ m in the 2000s one. Such a spatial pattern was reversed during the quiescent phases (Fig. 5b, f), when the lower tongue lost up to $104 \pm 4$ and $76 \pm 2$ m over 1980–2000 and 2003–2020, respectively. Meanwhile, the accumulation area gained up to $14 \pm 4$ and $12 \pm 2$ m, respectively, during these two periods of time, despite a consistent trend of overall mass loss on the whole glacier (Barandun et al., 2015; Denzinger et al., 2021). When considering only the terminus region (the lowest 0.6 km$^2$ before front advance, as delimited by Emelyianov et al., 1974), ice volume almost doubled during both active phases (Fig. 5d, h), and mean thickness over the same region increased in the 1970s from 57 m to 121 m and in the 2000s from 38 m to 67 m. Absolute values were larger in the 1970s; given that the advance covered a longer distance (Fig. 2), the considered terminus region (Sect. 2.3.3) is located further upstream from the snout than that in the early 2000s (Fig. 5a, e). Some subseasonal trends are also captured in Fig. 5g, h: downstream mass transfer in the summer of 2003 was stronger than ice ablation, leading to a volume increase during August, while the opposite is true for the year 2005. Ice thickness changes derived from SPOT 5 HRS closely match the ASTER results at corresponding dates, over both the terminus region and the broader tongue below transverse profile 4 (Fig. 5g, h). Uncertainties are generally smaller in HRS DEMs than in ASTER, reflecting the higher spatial resolution of the original data. Calculations on HRS DEM pairs over 5 day intervals (Sect. 2.2.2) yielded mean elevation changes between $-0.5 \pm 3.0$ and $0.1 \pm 1.6$ m over 67.5 km$^2$ of RGI polygons, confirming the stability of our results. The distribution of standardized elevation differences is very similar and close to Gaussian both on- and off-glacier (Fig. 6).

#### 3.3.2 Geodetic mass balance

Geodetic mass balance results are presented in Table 3 and Fig. 7a for six time intervals between 1973 and 2022. All periods were characterized by mass loss; the largest values correspond to the years immediately following the 1972–1973 pulsation and to the period 2020–2022. This last period displays exceptional thinning over the entire glacier extent up to the highest elevations (Fig. 7b). By contrast, we found a mass budget close to balanced between 1980 and 2003, concurrent with a front retreat of 950 m.



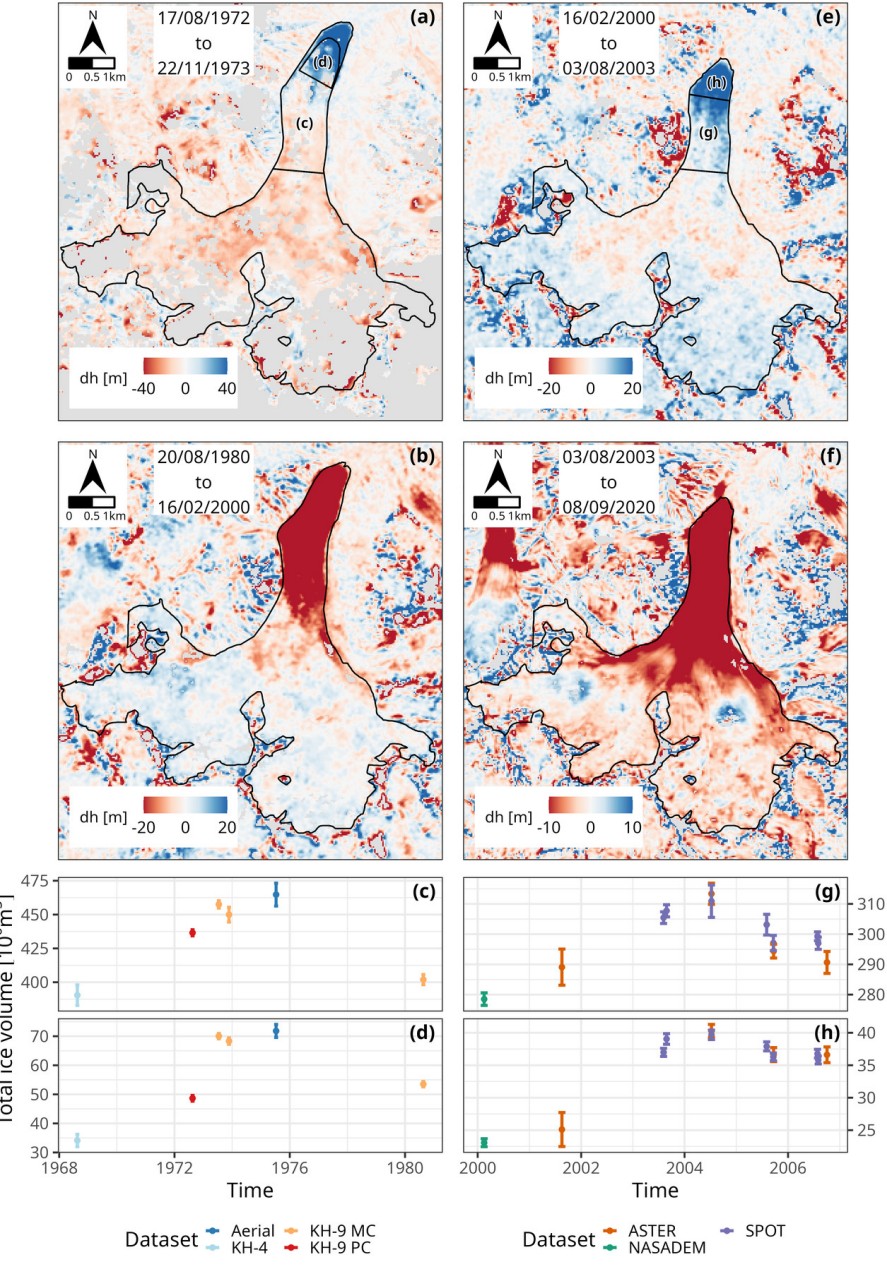

**Figure 5.** Abramov glacier thickness and volume changes. **(a, b, e, f)** DEM differences during two surge cycles. **(a)** Active phase of 1972–1973. **(b)** Quiescence over 1980–2000. **(c)** Active phase of 2000–2003. **(d)** Quiescence over 2003–2020. Color scale ranges are limited to preserve detail in the regions of slower change. Letter-marked polygons correspond to the ice volume time series of panels **(c, d, g, h)**. **(c, d)** Evolution of total ice volume over **(c)** the tongue below profile 4 and **(d)** the terminus region, from 1968 to 1980. **(g, h)** As **(c, d)** but from 2000 to 2006. Boundaries of the regions are shown in **(a)** and **(e)**. Error bars reflect only the uncertainty of surface DEMs, not the bed elevation uncertainty which affects systematically all values by a same amount.




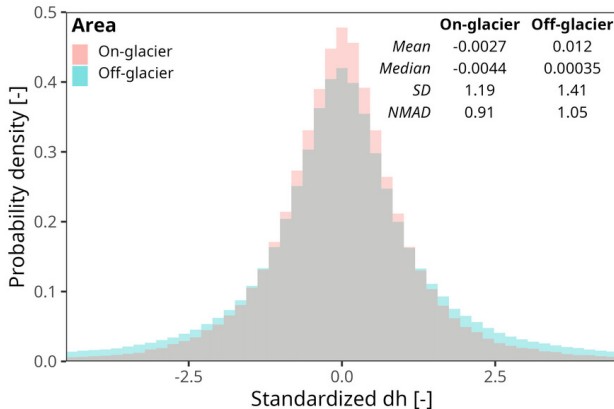

**Figure 6.** On- and off-glacier distribution of standardized elevation differences within HRS DEM pairs at 5 day intervals.

**Table 3.** Geodetic mass balance of Abramov glacier, including seasonality correction. Uncertainty ranges are discussed in Sect. 4.3. Reference initial volume (1973) is 2.55 km$^3$.

| Date | Specific annual mass balance [m w.e. yr$^{-1}$] | Cumulative volume loss [%] | Volume loss rate [% yr$^{-1}$] |
|---|---|---|---|
| 1973 | | | |
| | -0.89 ± 0.44 | 5.6 | 0.80 |
| 1980 | | | |
| | -0.10 ± 0.05 | 7.5 | 0.08 |
| 2003 | | | |
| | -0.27 ± 0.09 | 9.4 | 0.23 |
| 2011 | | | |
| | -0.50 ± 0.15 | 11.1 | 0.42 |
| 2015 | | | |
| | -0.39 ± 0.05 | 12.7 | 0.33 |
| 2020 | | | |
| | -1.05 ± 0.10 | 14.5 | 0.88 |
| 2022 | | | |

# 4 Discussion

## 4.1 Glacier dynamics

The compiled datasets provide information on Abramov glacier dynamics at high spatial and temporal resolution over 55 years.
In particular, frequent scene availability in the late 1990s and early 2000s enables reconstruction of glacier evolution during the period of interrupted *in situ* monitoring, including an unobserved phase of fast ice flow. Over more than five decades, we



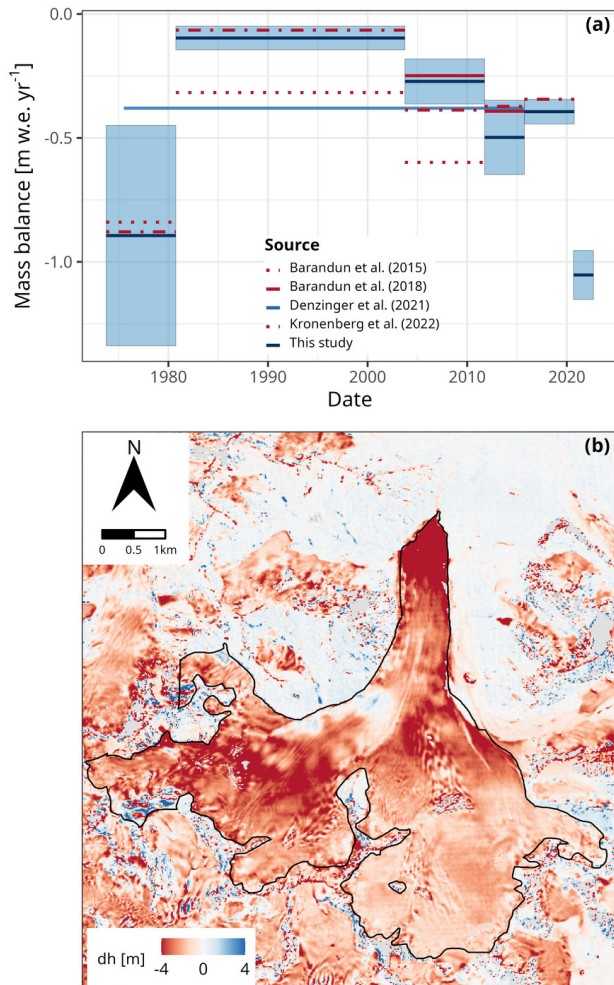

**Figure 7. (a)** Specific annual mass balance of Abramov glacier from our DEM differences and from previous studies; for readability, uncertainty is shown only for our results. **(b)** Pléiades DEM difference between 08/09/2020 and 05/09/2022.

found no direct relationship between trends of ice thickness and velocity (Fig. 3). Rather, ice thickness patterns in 1972–1973 and 2000–2003 (Fig. 5) provide evidence of rapid mass redistribution from an upper reservoir to the lower tongue, followed by longer periods of recovery in which surface height trends are inverted. These results indicate unstable flow conditions typical of surge-type glaciers (e.g., Raymond, 1987; Round et al., 2017; Benn et al., 2019), challenging the representativity of Abramov as reference for mass balance. Still, the whole region features a high density of surge-type glaciers (Glazirin' and Shchetinnikov, 1980; Goerlich et al., 2020; Guillet et al., 2022), which could put into question the representativity of even a stable glacier towards regional extrapolation of the observed trends of mass balance.



The heterogeneous character of the two pulsations (Sect. 3.2) suggests long-term evolution of the mechanisms behind ac-
celerated flow. The low and declining intensity of active phases could denote a progressive transition away from the unsteady
flow regime and towards more stable conditions, possibly as a result of multidecadal negative mass balance (Dowdeswell et al.,
1995; Denzinger et al., 2021). Enhanced ablation of the reservoir area in the most recent years (Fig. 7b) is expected to further
reduce mass buildup, depriving the glacier of enough ice to be transferred downstream in a surge (e.g., Flowers et al., 2011).
However, we also found the possible attainment of a new modest velocity peak by 2022/23 (Fig. 4c, d), about 20 years after
the previous one. If confirmed, a shorter duration of the quiescent phase (compared with the 30 years for the first observed
one) would be consistent with a long-term increase of net annual accumulation rates (Dowdeswell et al., 1995; Eisen et al.,
2001), which was quantified at about 50 % since the 1970s in the upper accumulation area (Kronenberg et al., 2021). Further
interpretation of dynamic instability at Abramov should involve ice flow modeling and an examination of glacier dynamics at
the mountain range scale.

## 4.2  Comparison with *in situ* observations and past studies

Historical variations of glacier length, thickness, and velocity until the 1990s have been described in numerous Soviet and
Russian publications based on *in situ* measurements (Emelyianov et al., 1974; Abulkhasanova et al., 1979; Suslov et al.,
1980; Dolgoushin and Osipova, 1982; Bassin and Kamnyansky, 1987; Glazirin' et al., 1993). A major goal of our study is to
extend these time series, bridging the gap in the observations. However, rigorous quantitative comparison with our results is
not straightforward due to several factors. First, spatial delimitation and aggregation of measurements is not always clear in
the cited publications as, apparently, reported length changes often refer to the lowest point of the glacier and not to width
averages as in our results, and whether stagnant debris-covered ice is considered part of the glacier or not remains unknown.
The definition of "terminus region" and "glacier tongue" is also sometimes vague and inconsistent across authors. Second,
reported time intervals of the changes are often imprecise, usually omitting the day specification – introducing a potential
location mismatch by several tens of meters during episodes of fast flow. Third, even well-specified time intervals do not
exactly match the available dates of our scenes. Fourth, we find considerable disagreement in the numbers reported by different
sources, even in relation to a single *a priori* well-defined variable such as total advance of the glacier snout over the early
1970s: 420 m (Emelyianov et al., 1974), 500 m (Abulkhasanova et al., 1979), 401 m (Suslov et al., 1980), 625 m (Dolgoushin
and Osipova, 1982), 400 m (Bassin and Kamnyansky, 1987). Thus, comparisons against our remote sensing measurements –
including front variations and ice volume changes – are potentially affected by inconsistent reporting of the *in situ* values.

For glacier length, our results (Fig. 2) indicate a progression by $350 \pm 6$ m between 17 August 1972 and 22 November 1973,
when the terminus had already reached the maximum extent within a few meters. Combined with an *in situ* observation of ad-
vance by 82 m from September 1971 to November 1972 (Suslov et al., 1980), our observations support the lower values (about
400 m) among the range of Soviet estimates of total advance presented above. A recent reconstruction of Abramov glacier
terminus positions was presented by Mandychev et al. (2017), also based on remote sensing and topographic data. It has no
overlap with our datasets: the study used the Landsat archive, some unrectified aerial frames which were not available to us,
and several schematic maps of the glacier margin digitized from Soviet publications. The comparison (Fig. 8) shows a qualita-



tive agreement of the overall trends of retreat and pulsations, but also several differences, likely related to both residual image distortions and lower scene resolution within that study. In several periods, the glacier is incorrectly classified as advancing – of note, 6 years between 1977 and 1984, up to a 1978–1980 rate of 46.5 m yr$^{-1}$. We find these results to be implausible, as they would lead to a terminus position 100 m forward in 1984 compared with the maximum of 1975 – in contrast with both the observed terminal moraine position and the very low ice velocity during the period (Fig. 3). During the more recent pulsation, Mandychev et al. (2017) report terminus advance starting in 2000 instead of 2002. This is apparently based on the omission of debris-covered ice at the terminus, visible on our SPOT and IRS images but not on lower-resolution Landsat scenes.

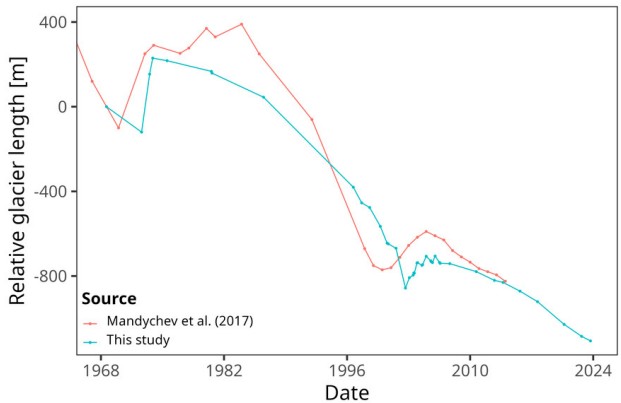

**Figure 8.** Length change of Abramov glacier from Mandychev et al. (2017) and from our study.

With the caveats mentioned above, we compare our estimates of ice thickness and volume changes against the *in situ* observations of the 1970s. For the net ice volume transferred to the tongue over 1972–1973, Suslov et al. (1980) report a value of $23.12 \cdot 10^6$ m$^3$, calculated from measurements of ice velocity and transverse cross-section (at profile 4) and surface ablation (downstream of profile 4). The DEM difference from KH-9 PC and MC (Fig. 5a) yields a total change of $32 \pm 10 \cdot 10^6$ m$^3$ over the same region between 17 August 1972 and 22 August 1973. Our measurement includes the whole period of fastest flow but misses the ice ablation of early summer 1972. For the 0.6 km$^2$ region of interest covering the terminus (Fig. 5a), Emelyianov et al. (1974) report a doubling of the total ice volume over the first 8 months of 1973. The value is slightly larger than the change that we obtain in Fig. 5d, but within the error range when considering uncertainty of the glacier bed elevation. Due to systematically incomplete coverage of our DEMs in the accumulation area, we cannot quantify the total mass transferred away from the reservoir area over the early 1970s surge. However, our estimated thinning by up to $20 \pm 5$ m (Fig. 5a) is very close to the values reported for the firn area cross-profiles in Suslov et al. (1980) (16.5 m) and on the map in Kotlyakov (1997) (20 m). Thus, we conclude that our estimates corroborate *in situ* measurements within the respective uncertainty ranges, despite the lack of rigorous comparison.

For mass balance (Fig. 7a), our results generally agree well with past estimates. Our results match within uncertainty the geodetic mass changes calculated over 1975–2015 by Denzinger et al. (2021) from structure-from-motion (SfM) and Pléiades



data. Geodetic mass balance including two SPOT DEMs from 2003 and 2011 was also presented by Barandun et al. (2018) as
validation of a daily mass balance model: our value for that period is predictably very similar (within 0.1 m w.e. $yr^{-1}$), but we
obtain a DEM with more complete coverage (97 % vs 74 %). The mean geodetic mass balance computed for 1973–1980 (Table
3) matches historical glaciological estimates within 0.1 m w.e. $yr^{-1}$ (Glazirin' et al., 1993; Pertziger, 1996; Kamnyansky,
2001). More recent periods can be compared against the calibrated model reanalyses of Barandun et al. (2015), Barandun
et al. (2018), and Kronenberg et al. (2022). Our results generally support the findings of the full energy-balance simulation
in Kronenberg et al. (2022), except over 2003–2011, where glaciological calibration data are missing. For that period, the
temperature-index approach in Barandun et al. (2018) – which is constrained by transient snow line elevation from Landsat –
is the most similar.

Additional comparisons with remote sensing studies covering Abramov glacier are provided in Appendix E. In particular,
the active phase of the early 2000s is not well resolved by Landsat or ASTER, with regard to both surface ice movement on
the tongue (Fig. E1a) and ice thickness trends at the terminus (Fig. E2).

## 4.3 Measurement uncertainties

Throughout the present study we quantify uncertainties with common methods mostly based on the analysis of stable terrain
in the glacier vicinity. Comparison with *in situ* measurements from the 1970s indicates reasonable estimates for our observa-
tions and their uncertainties (Sect. 4.2). Still, lack of ground validation over the early 2000s is a significant limitation. In the
following, we discuss some sources of uncertainty affecting our results.

The interpretation of glacier margins under debris or snow cover can introduce major errors in estimated glacier area and
length changes (Paul et al., 2013, 2017, 2020). However, the achieved high temporal frequency of orthoimages and concurrent
DEM differences allows close tracking of surface height and morphology changes, enabling a consistent interpretation of
glacier extents (e.g., Racoviteanu et al., 2009; Paul et al., 2020).

Our estimates of ice velocity uncertainty (Fig. 4a) mostly reflect the residual geometric inaccuracies within image pairs;
still, error sources specifically affecting ice velocities are not that easily accounted for (e.g., Altena et al., 2022; Zheng et al.,
2023). In particular, more frequent and severe mismatches are observed on-glacier, due to repeat formation of self-similar
features such as ogives, crevasses, seracs, and avalanche deposits (Paul et al., 2015). Also, the melting surface can lead to
more decorrelation and a different noise distribution compared with stable terrain (Millan et al., 2019; Zheng et al., 2023).
Indeed, noise along the profiles in Fig. 4c, d is visibly larger than our NMAD estimate of 0.5 to 1.5 m $yr^{-1}$ (Sect. 3.2). We
also notice the potential impact of seasonal velocity changes on the Sentinel-2 velocity mosaics: the contributing image pairs
(and thus the covered time intervals) are in principle different at each location, potentially leading to biases depending on the
relative representation of the various seasons. A better characterization of uncertainties would require ground-based long-term
measurement of annual and seasonal velocities. In one comparison with GPS over the Mont-Blanc region, Millan et al. (2019)
found annual discrepancies of Sentinel-2 mosaics by 4–6 m $yr^{-1}$ for glaciers of width and velocity similar to those of Abramov.

The statistical framework of Hugonnet et al. (2022) allows uncertainty quantification in DEM differences from stable terrain,
including the effects of heteroscedasticity and spatial correlation. Here, we also validate the results with the empirical spatial



sampling in Berthier et al. (2016) and Miles et al. (2018). Still, our thickness change results over 1980–2000 and 2000–2003
(Fig. 5b, e) involve NASADEM heights affected by C-band radar penetration of dry snow and ice, which can lead to systematic
uncertainties (especially in upper accumulation areas) which may not be fully represented on stable terrain (Gardelle et al.,
2012; Barandun et al., 2015). Correction of the possibly under-estimated glacier surface height for 2000 is not straightforward.
In general, we would expect an intensification of the patterns of firn area thickening over 1980–2000 and thinning in the
later period – magnitude of surge-like patterns is possibly under-estimated by the use of NASADEM. For simplicity, here we
assume that radar penetration involves just the layer of seasonal snow, returning the height of the (mostly thinly debris-covered)
summer glacier surface (Buckley et al., 2023). Such an assumption probably becomes inadequate over the upper reaches of
the accumulation area, well above our regions of interest (Bannwart et al., 2024). The magnitude of observed thickness change
patterns (Sect. 3.3.1) exceeds the expected C-band penetration by a factor of two or more (Li et al., 2021; Fan et al., 2023).

For geodetic mass balance, the error budget is in most cases dominated by surface height change uncertainty; over 1973–
1980, this is especially large (Fig. 7a) as a result of numerous data gaps in the accumulation area (Sect. 2.3.3). Of all DEM
differences, the 2020–2022 Pléiades map (Fig. 7b) is the most accurate, with an aggregated error of just $\pm$ 0.1 m over the
Abramov outline and almost no gaps or seasonal mismatch. Glaciological mass balance is in excellent agreement, with a
2020–2022 mean value of $-1.03$ m w.e. yr$^{-1}$ (World Glacier Monitoring Service (WGMS), 2024). However, the short obser-
vation interval renders density conversion highly uncertain, and the 60 kg m$^{-3}$ error assumption probably becomes inadequate
(Huss, 2013). In particular, exceptional thinning above the firn line could be buffered by firn density increase at depth due to
enhanced meltwater refreezing, leading to a more positive internal mass balance and thus systematic over-estimation of mass
loss (Kronenberg et al., 2022; Berthier et al., 2023). Still, Abramov glacier has a mostly temperate firn area with decreasing
pore space over the last few decades (Kislov et al., 1977; Kronenberg et al., 2022), thus limiting the potential for meltwater
refreezing. A conservative lower bound for the total change can be computed assuming zero balance over the 8 km$^2$ of snow-
and firn-covered area. The result ($-0.80$ m w.e. yr$^{-1}$) is still exceptionally negative compared with that in all subperiods since
1980.

## 4.4 Evaluation of datasets and methods

Declassified satellite scenes have been increasingly used to assess historical glacier evolution since the 1960s to 1980s, both
at regional to global scale (e.g., Pieczonka and Bolch, 2015; Maurer et al., 2019; Zhou et al., 2019; Dehecq et al., 2020;
Bhattacharya et al., 2021) and for individual catchments and large glaciers (e.g., Bolch et al., 2011, 2017; Goerlich et al.,
2017; Ghuffar et al., 2023). However, these studies usually lack ground validation during the early periods, especially for the
data-scarce Central Asia region. In the present work we observe that length and volume changes from KH-4 and KH-9 are
consistent with *in situ* measurements over one surge cycle of the relatively small Abramov glacier. Moreover, accuracy derived
from stable terrain is similar to a 1975 SfM DEM obtained from a dedicated aerial survey (Fig. 5c, d).

From the SPOT, IRS, and RapidEye archives we assembled a dataset of late-summer imagery at 5 m, mostly complete at
annual frequency between the mid-1990s and the late 2010s. Our time series successfully bridges the gap in ice dynamics
observations at Abramov. In fact, the covered time span is of particular interest for all reference glaciers in Central Asia, whose





monitoring was interrupted following the Soviet Union collapse, and for sites which have been newly established over the last few years and lack detailed historical observations, for example in Tajikistan (Barandun et al., 2020). Within our dataset we

find an especially dense scene availability during the early 2000s (Table 2; Fig. 2b). Between 2000 and 2005, we observe more than twice as many cloud-free summer scenes than from Landsat and ASTER combined, owing to irregular acquisitions by the latter platforms over their initial period of operation. Thus, SPOT and IRS have the potential to become a primary data source to reconstruct glacier change over those years.

Beside scene availability, the higher spatial resolution unlocks details of ice dynamics which are poorly resolved on Landsat

and ASTER (Fig. 5, Fig. 8; Appendix E). Indeed, dynamics of smaller and slower-flowing glaciers are beyond reach of optical remote sensing from such sensors (Paul et al., 2022). A notable consequence is that most inventories of surge-type glaciers can be biased towards larger entities (Goerlich et al., 2020). Higher-resolution imagery can support the assessment of unstable flow at the smaller ice bodies.

To derive ice velocities, we performed several mixed correlations involving scenes from different sensors (SPOT with IRS,

RapidEye with Pléiades; Table A1). To the best of our knowledge, such multisensor correlation has not been previously applied on glaciers. Thanks to full control over the orthorectification process (unlike with Landsat and Sentinel), the resulting displacement maps show limited residual distortions (Fig. 4a).

Several studies have used different spectral bands of Sentinel-2 to derive glacier surface displacements (Altena and Leinss, 2022). Results from the various bands usually match within uncertainty ranges, indicating adequate interband registration and

calibration (Lacroix et al., 2018; Altena et al., 2022). Still, in our four-band composites, we notice a small noise reduction in the upper accumulation area compared with velocities derived from a single band; we observe virtually no change on the glacier tongue.

Overall, the methods used throughout the study are mainly based on open-source tools. Most steps do not require manual intervention, except for (1) cloud-free image selection (for which some automated approaches already exist: e.g., Fisher, 2014),

(2) front digitization, and (3) manual picking of initial GCPs (in case an appropriate reference orthoimage is not already available for their automated detection). For GCPs-based orthorectification, we consider the availability of a high-resolution reference such as Pléiades not mandatory: a cloud-free scene from Sentinel-2 can be used instead. Thus, application at a larger scale could be relatively efficient to derive time series of regional-scale velocity and thickness changes for the early 2000s.

## 5 Conclusions

In the present study we compile a 1968–2023 dataset of high-resolution orthoimages and DEMs at Abramov glacier by processing raw optical scenes from multiple archives. Our reconstruction allows studying the long-term evolution of unstable ice dynamics, revealing two cycles of mass redistribution typical of surge-type glaciers. During quiescence, we found a sustained gradual acceleration, quite uniform over the glacier tongue. The observed phases of fast flow show decreasing intensity and increasing duration, possibly suggestive of a multidecadal transition towards more stable dynamics. We also note the potential

attainment of a third modest peak of ice velocity by 2023.



Our observations are in good agreement with results from historical and modern research at the site. We find that length and volume estimates from declassified reconnaissance satellites are consistent with *in situ* measurements. The 5 m scenes of IRS, SPOT, and RapidEye extend the observational time series of ice dynamics at Abramov, bridging the 12 year gap of interrupted monitoring. In particular, they allow quantification at subseasonal scale of the second unobserved active phase, which is poorly

resolved by Landsat and ASTER products. The 1973–2022 geodetic mass balances over six subperiods support the negative estimates of recent model-based studies.

Our methods can be expanded to a larger scale efficiently to investigate regional evolution of glacier velocity over the past two decades. Such work would extend early studies on the distribution of unstable ice flow in the Pamir-Alay (Glazirin' and Shchetinnikov, 1980; Glazirin' et al., 1987). Compared with the Landsat record, we anticipate a better coverage of ice

dynamics, especially on smaller and slower-moving entities, supporting compilation of more complete inventories of surge-type glaciers. At Abramov glacier, further interpretation of the observed dynamics should involve ice flow modeling, constrained by the velocity and volume changes derived within our study. Of particular interest would be the relationship between the observed unstable flow dynamics and the well-known mass balance of this reference glacier.

*Code and data availability.* Raw declassified scenes can be accessed at the EarthExplorer (https://earthexplorer.usgs.gov/). DEMs and or-

thoimages from KH-9 MC, SPOT, and Pléiades will be made available upon publication. IRS and RapidEye scenes should be requested respectively from the Antrix Corporation and Planet Labs. Digitized terminus positions will be submitted to the database of the World Glacier Monitoring Service. Software pipelines for scene orthorectification, velocity derivation and postprocessing, and analysis of terminus positions and DEM differences are available at https://doi.org/10.5281/zenodo.12731407. Most Russian-language publications cited within the study can be accessed through https://sites.google.com/view/glaciobiblio/.

## Appendix A:  Lists of satellite scenes

Here, we provide the list (Table A1) of orthoimages used for surface velocity derivation (Fig. 3) and the list (Table A2) of processed declassified scenes from reconnaissance satellites.

## Appendix B:  Orthorectification of SPOT scenes

Our workflow for SPOT orthorectification is closely based on the geoCosiCorr3D pipeline of Aati et al. (2022). The main

difference is the use of NASA ASP tools (instead of MicMac) for tie point detection between the raw and reference images. We adopted this change because of the observed inability of MicMac to obtain a sufficient number of tie points between Pléiades and SPOT scenes. Thus, we initially derived tie points by using NASA ASP tools *ipfind* and *ipmatch* between the raw SPOT images and the Pléiades 2020 reference, downsampled to match the SPOT resolution (Table 2). However, monoscopic SPOT scenes have a $60 \times 60$ km footprint which extends significantly beyond the Pléiades coverage, as well as highly variable cloud

and snow cover. These traits complicated the automatic detection of sufficient well-distributed GCPs from the Pléiades scene,



**Table A1.** Orthoimages at 5 m resolution used for single-pair annual surface velocity derivation.

| Date | Platform | Notes |
|------|----------|-------|
| 15/09/1996 | SPOT 3 | PAN, resampled from 10 m |
| 30/08/1997 | IRS-1C | PAN |
| 28/07/1998 | IRS-1D | " |
| 02/09/2000 | IRS-1C | " |
| 02/09/2001 | IRS-1C | " |
| 23/08/2002 | IRS-1C | " |
| 27/08/2003 | SPOT 5 | " |
| 12/08/2004 | IRS-1C | " |
| 03/08/2005 | SPOT 5 | " |
| 02/08/2006 | SPOT 5 | " |
| 06/09/2007 | SPOT 5 | " |
| 11/09/2010 | RapidEye | Band 2 |
| 08/09/2011 | RapidEye | " |
| 01/08/2012 | RapidEye | " |
| 07/09/2013 | RapidEye | " |
| 02/09/2014 | RapidEye | " |
| 20/08/2015 | Pléiades | PAN, resampled from 0.5 m |
| 27/07/2016 | RapidEye | Band 2 |
| 31/08/2017 | RapidEye | " |
| 05/09/2018 | RapidEye | " |
| 26/07/2019 | RapidEye | " |
| 08/09/2020 | Pléiades | PAN, resampled from 0.5 m |

limiting the accuracy of the result or even preventing successful orthorectification in some cases. Thus, with the Pléiades reference, we orthorectified just two cloud-free scenes from SPOT 5 HRG, collected in 2006: a winter one with favorable illumination conditions and a late-summer one with similar snow cover to that from Pléiades. We then adopted the resulting orthoimages as reference for all other (winter and summer) SPOT scenes. The ample overlap enabled GCP detection even with

challenging snow and cloud cover conditions. After matching the tie points, we discarded those falling close to unstable terrain or to the scene margins (since patch correlation is used in geoCosiCorr3D for RSM refinement). In most cases, this selection still left several hundreds unevenly distributed tie points. To optimize both spatial coverage and processing performance, we limited the number of matches to 60 for each scene. After random selection of the first point, we iteratively selected the next one based on maximization of the average horizontal distance to the already-selected points. This ensured complete and uniform

spatial coverage of each scene (Fig. B1). We converted tie points to GCPs by extracting their elevation from NASADEM.



**Table A2.** Processed declassified scenes from reconnaissance satellites.

| Date | Sensor | Id | Notes |
|------|--------|-----|-------|
| 18/08/1968 | KH-4 PC | DS1104-2169DA097 | |
| 17/08/1972 | KH-9 PC | D3C1203-400520A048 | |
| | | D3C1203-400520F048 | |
| | | DS1104-2169DF091 | |
| 16/07/1973 | KH-9 MC | DZB1206-500007L015001 | |
| | | DZB1206-500007L016001 | |
| 22/11/1973 | KH-9 MC | DZB1207-500041L001001 | Tristereo |
| | | DZB1207-500041L002001 | |
| | | DZB1207-500041L003001 | |
| 11/06/1975 | KH-9 PC | D3C1210-100030A111 | |
| | | D3C1210-100030F111 | |
| 25/07/1980 | KH-9 PC | D3C1216-200279A037 | |
| | | D3C1216-200279F036 | |
| 20/08/1980 | KH-9 MC | DZB1216-500273L006001 | Tristereo |
| | | DZB1216-500273L007001 | |
| | | DZB1216-500273L008001 | |

Finally, we processed all sets of scenes and GCPs with the RSM refinement and orthorectification of geoCosiCorr3D (Aati et al., 2022).

We observed a rather strong dependence of the GCP spatial distribution (and thus orthorectification result) on the parameters of *ipfind* and *ipmatch*: $N$, the target number of feature points to be detected, and $T_m$, the threshold for RANdom SAmple Consensus (RANSAC) of point matching, respectively. Specifically, the value of $N$ must be sufficiently large to enable detection of enough tie points, even on scenes with large snow cover or small overlap to the reference. At the same time, a very large $N$ and small $T_m$ tended to produce GCPs clustered within narrow subregions of the images, with a low matching error but incomplete spatial coverage, and subsequently inaccurate orthorectification. Finally, a large value of $T_m$ sometimes failed to exclude wrongly matched tie points, also leading to imprecise orthorectification. Thus, we ran the pipeline with multiple values of $N$ and $T_m$ (within $[10^5, 10^7]$ and $[10, 100]$, respectively); for each scene, we selected the parameter combination with the lowest residual error among those with full GCP coverage of the overlap region between raw and reference scene.

For orthorectification of all monoscopic scenes, including SPOT, we used the NASADEM topography as reference terrain. Such a single DEM does not reflect surface height change over the years, potentially leading to horizontal biases in the resulting glacier imagery (Kääb et al., 2016). In Sect. 2.2.2, we generate SPOT 5 HRS DEMs, which in several cases match the date of monoscopic HRG scenes, providing a simultaneous terrain surface for orthorectification. However, such DEMs have inconsistent coverage and present gaps and errors from pixel saturation and stereo matching blunders. Thus, using them for




orthorectification would involve nontrivial mosaicking and interpolation, again leading to orthophoto distortion. Moreover, the Abramov glacier tongue (where height change is largest) follows an overall NNE orientation of 14°, which matches within ± 1° the local ground track of sun-synchronous satellites (Capderou, 2005). In other words, ice flow is parallel to the satellite

track, while horizontal biases introduced by an inaccurate DEM mainly affect the cross-track direction for nadir acquisitions (Kääb et al., 2016). As such, we expect only minimal impact on our derived surface velocities. We verified this conclusion by testing orthorectification with the Copernicus GLO-30 terrain instead of NASADEM (average 13 ± 5 m height difference over the glacier tongue). Over 2005–2006 (Fig. 4a), the bias of derived velocity is between −0.2 and +1.0 %, well within uncertainty range.

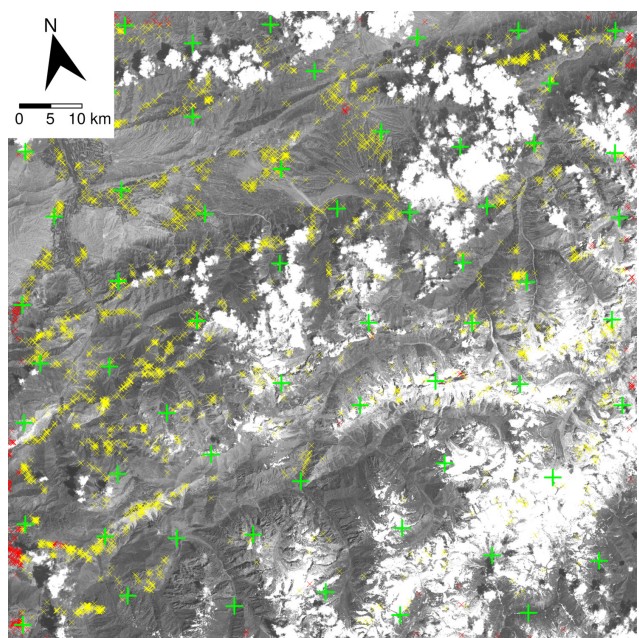

**Figure B1.** Raw SPOT 5 HRG scene from 03/08/2005. Green marks are the final selection of GCPs for orthorectification; red and yellow indicate, respectively, discarded and excess tie points.

**Appendix C:  SPOT DEM generation**

We calculated SPOT DEMs with NASA ASP using the GLO-90 DEM as initial terrain guess. We also tested GLO-30 but found the occurrence of ghosting artifacts in the output when generated from such a higher-resolution seed (Beyer et al., 2023, page 32). The 8-bit HRS scenes commonly suffer from pixel saturation over snow-covered areas (Barrou Dumont et al., 2024), in principle leading to data gaps on glacier accumulation areas in summer. However, we observed a tendency of the

MGM algorithm to compute height values even in fully saturated areas, due to interpolation of voids in the disparity maps (Dehecq et al., 2020). Thus, we postprocessed each HRS DEM by discarding all pixels corresponding to saturated digital





numbers in the respective input scenes. We subsequently coregistered the full DEMs to NASADEM with the approach in
Nuth and Kääb (2011) followed by a 2D tilt correction (Girod et al., 2017), using RGI outlines as unstable terrain. In four
cases we found Abramov glacier to be split over two overlapping stereo pairs, acquired at a few minutes of interval. In the
corresponding DEMs, we observed seam artifacts at the edges, preventing a straightforward mosaicking. We corrected the
artifacts by introducing a second step of fine local coregistration of these split DEMs to NASADEM, applied only within a
reduced region around Abramov. Because of increased terrain ruggedness and instability in this mountainous region compared
with the full HRS scenes, we used a more restrictive definition of stable terrain: we excluded a 200 m buffer around the
RGI outlines, all slopes greater than 30° (Denzinger et al., 2021), and all cells with an elevation difference greater than four
NMADs before coregistration. Then we mosaicked the coregistered DEM pairs, using simple arithmetic average over the
regions of overlap. In all cases the seam artifact at Abramov disappeared. In order to obtain a DEM with full glacier coverage,
we also combined two short-interval HRS DEMs, from 1 and 6 August 2003. The former was acquired at a low gain setting,
preserving details over the snow-covered accumulation area, while the latter provides cloud-free coverage of the lower tongue:
the combination achieved a DEM with elevation data on more than 99.9 % of the glacier extent, that we used in ice thickness
and volume comparisons (Fig. 5 and 7a).

**Appendix D:  Flow continuity**

We checked consistency of our estimated ice velocities and volume changes by means of the continuity equation applied to the
terminus region of the glacier (Fig. 5e). Assuming no change in bed separation, ice influx to the terminus can be expressed in
discrete form, integrated over both time and space (Reynaud et al., 1986; van der Veen, 2013):

$$Q = \left( \overline{\Delta H} - \frac{\overline{B}}{\rho_i} \right) \cdot A \; = \; k \Delta t \sum_{i=1}^{N} u_{s,i} H_i L_i \qquad \text{(D1)}$$

where $\overline{\Delta H}$ is the surface height change and $\overline{B}$ the surface mass balance, both averaged over the terminus region of area $A$
and measured over period $\Delta t$; $\rho_i$ is ice density; summation is performed over the $i = 1, ..., N$ segments of a transverse flux
gate at the upper boundary of the terminus region; $u_{s,i}$ is the surface ice velocity perpendicular to the flux gate, $H_i$ is the
corresponding ice thickness, $L_i$ is the length of the transverse segment, and $k$ is the mean ratio between depth-averaged and
surface ice velocity. We derived $\overline{\Delta H}$ from DEM differentiation (Sect. 2.3.3), $\overline{B}$ from the daily model of Kronenberg et al.
(2022), $u_i$ from the ice velocities of Sect. 2.3.2, and $H_i$ from the bed map of Kuz'michenok et al. (1992). Thus, in Eq. D1,
the ratio $k$ remains the only unknown parameter. We computed this by inverting the equation over the subperiods for which
all required variables were simultaneously available: the three annual intervals between 6 August 2003, 9 July 2004, 3 August
2005, and 2 August 2006. We computed uncertainties in our remote sensing data as per Sect. 2.3.2 and 2.3.3; in the surface
mass balance, from the RMSE between the model of Kronenberg et al. (2022) and the glaciological measurements (0.7 m w.e.
yr$^{-1}$); in the ice thicknesses, with a conservative ± 20 % based on literature values (e.g., Grab et al., 2021); and in the ratio
$k$, with standard error propagation. From the left-hand side of Eq. D1, we derived ice influx values of 5.4 ± 0.8, 2.3 ± 0.9,



and $1.7 \pm 0.8$ million $m^3$ respectively for 2003–2004, 2004–2005, and 2005–2006. The numbers refer to the terminus region downstream of the lower cross-section in Fig. 5e. To match those results using the right-hand side of Eq. D1, we obtained

the following ratios of depth-averaged to surface ice velocity: $0.95 \pm 0.26$, $0.67 \pm 0.31$, and $0.67 \pm 0.38$. Despite the large uncertainties, results are all within physical ranges, confirming the consistency of our datasets. The high value for 2003–2004 corresponds to a dominant contribution of basal sliding, which brings depth-averaged velocity close to surface values (Cuffey and Paterson, 2010). Over the later years, the result is somewhat lower – but well within uncertainty range – when compared with the theoretical prediction of a simple shear profile: 0.8 in case of no basal sliding and 1.0 for slip-dominated

motion (Cuffey and Paterson, 2010). Our value of 0.67 is also very close to the ratio (0.64) proposed by Suslov et al. (1980) at Abramov for the 1960s and 1970s before the first pulsation, reportedly based on Palgov (1958). Unfortunately, the data and reasoning behind such an estimate appear to never have been published – thus, we do not speculate further as to potential interpretations.

## Appendix E:  Abramov glacier dynamics within global remote sensing datasets

We compared our findings against several published remote sensing studies, providing ice velocity and surface height changes at global scale. While some features of the Abramov dynamics are recognizable within the latter (Fig. 5g, h), our results compare favorably with existing datasets.

For annual surface velocity, we evaluated the velocities of ITS_LIVE, which covers 1985–2022 from Landsat 4–8 (Gardner et al., 2018, 2024), as well as the 2014–2021 Sentinel-1 grids of Friedl et al. (2021) and the 2017/18 biannual result of Millan

et al. (2022), based on Sentinel-1, Sentinel-2, and Landsat 8. The coarse grids of ITS_LIVE and Friedl et al. (2021) (at 240 and 200 m resolution, respectively) do not capture most of the surface displacement over the lower tongue (Fig. E1b, c). Of note, no velocity change is recognizable during the active phase of 2000–2003 (Fig. E1a). The result in Millan et al. (2022) is more detailed but includes several patches of outlier values and stronger noise in the flow direction (Fig. E1d, Fig. 4b).

For ice thickness patterns, we examined the 5 year grids of surface height change rate in Hugonnet et al. (2021) calculated at

100 m resolution from ASTER stereo pairs via Gaussian process regression (Fig. E2). We found that thickening at the terminus is not visible over 2000–2005, which would prevent an automated identification of the active phase from the dataset.

*Author contributions.*  EM designed the study, performed the analysis, and led the writing with contributions from all authors. EB processed the Pléiades data. AD processed the KH-9 MC data. TB, AB, and SG processed the KH-4 and KH-9 PC data. MB and MH provided data from previous studies at Abramov glacier. All authors contributed to the interpretation of the results.

*Competing interests.*  Some authors are members of the editorial board of journal The Cryosphere.



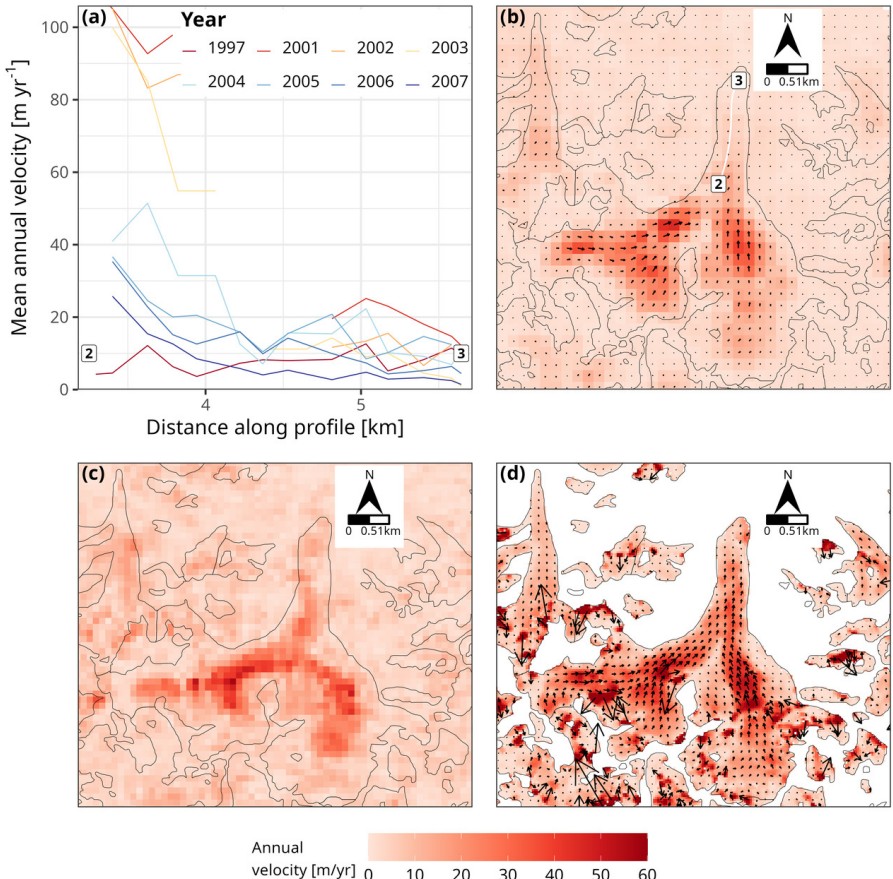

**Figure E1. (a)** Longitudinal profiles of mean annual surface velocity from ITS_LIVE (Gardner et al., 2024) during the second active phase. Profile geometry is shown in panel **(b)** and is the same as in Fig. 4b. **(b–d)** Maps of mean annual surface velocity in 2017/18 (compare with Fig. 4b) from **(b)** ITS_LIVE, **(c)** Friedl et al. (2021), and **(d)** Millan et al. (2022). For readability, the vector field of flow velocity is sampled (with arrows) at different intervals across the plots: original spatial resolution of the products is 240, 200, and 50 m, respectively.

*Acknowledgements.* We thank the project "Strengthening the resilience of Central Asian countries by enabling regional cooperation to assess glacio-nival systems to develop integrated methods for sustainable development and adaptation to climate change" funded by the Global Environment Facility / United Nations Development Programme / United Nations Educational, Scientific and cultural Organization (GEF/UNDP/UNESCO, contract no. 4500484501) and the project "Cryospheric Observation and Modelling for Improved Adaptation in Central Asia (CROMO-ADAPT)" (contract no. 81072443) funded by the Swiss Agency for Development and Cooperation and the University of Fribourg and the project "From ice to microorganisms and humans: Toward an interdisciplinary understanding of climate change impacts on the Third Pole (PAMIR)" (grant number: SPI-FLAG-2021-001) funded by the SPI Flagship Initiative of the Swiss Polar Institute. We gratefully acknowledge Tomas Saks and Esther Blokbergen for providing access to Russian-language literature. We thank Romain Hugonnet and Fanny Brun for the constructive discussions on DEM analysis, Horst Machguth and Alessandro Cicoira for the exchanges on ice dynamics, Joaquin Munoz Cobo Belart and Oleg Alexandrov for support with DEM generation, and Saif Aati for the valuable support





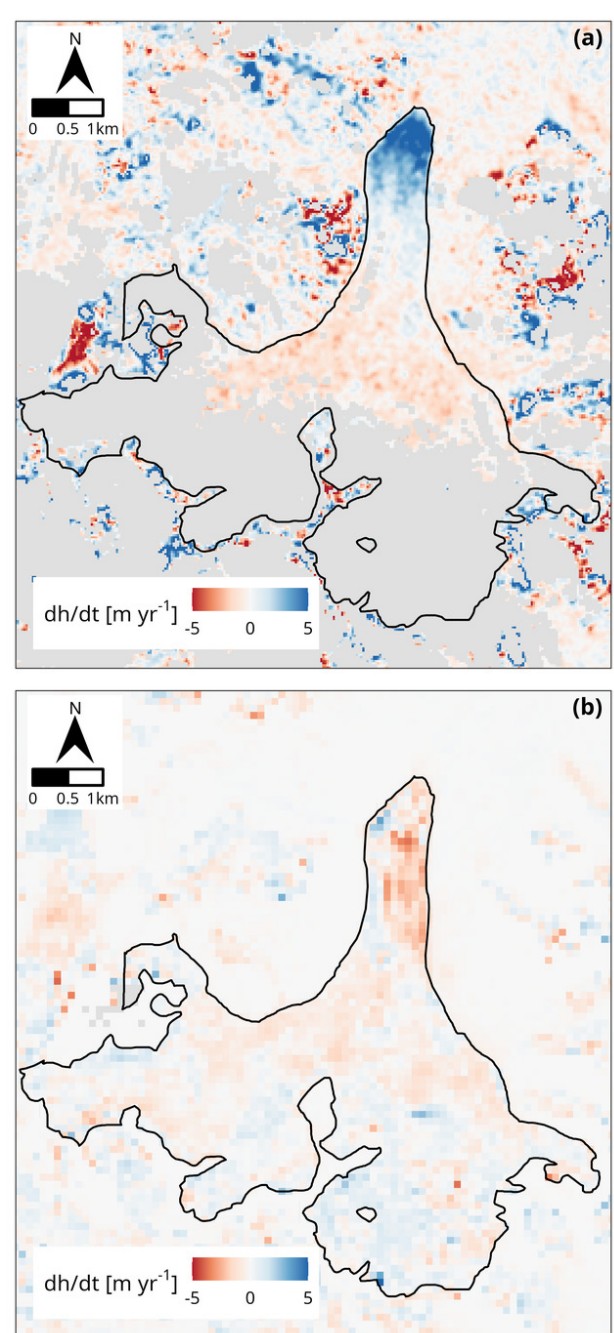

**Figure E2.** Annual surface height change rate over 2000–2005. **(a)** Our result, from NASADEM and SPOT. **(b)** Hugonnet et al. (2021), from ASTER.





on geoCosiCorr3D application. We would like to thank the following institutions: the Central Asian Institute of Applied Geosciences (CA-IAG), particularly Erlan Azisov, Ruslan Kenzhebaev, and Ryskul Usubaliev, as well as Kyrgyz Hydromet, particularly Sultan Belekov, for enabling and supporting the work on Abramov glacier. We acknowledge the use of SPOT images acquired by CNES's Spot World Heritage Programme, as well as the Education and Research Program of Planet Labs for access to the RapidEye scenes. EB acknowledges support from the French Space Agency (CNES). AB acknowledges the research funding (grant no. CRG/2021/002450) received from Science & Engineering Research Board (SERB), Department of Science & Technology (DST), India, under Core Research Grant (CRG).




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
