# Peer review of "Five decades of Abramov glacier dynamics reconstructed with multi-sensor optical remote sensing"

_EGUsphere, 2024_

## Referee Comment (RC2)

**Overall comments:**

The authors compile a 50+ year dataset of kinematic changes of Abramov glacier, filling in gaps in the *in situ* observational record using a variety of remote sensing datasets. Overall, the manuscript is well-written and demonstrates how more detailed datasets of glacier kinematics can reveal novel dynamic behavior that may complicate mass balance studies. I applaud the authors for the thoroughness of their data processing and presentation of the methodology. I recommend a minor revision of the manuscript with the specific comments listed point by point below.

**Specific comments:**

Abstract:
- L9: It would be helpful to mention which "archives" are used in this study, especially since your results show a newly-discovered pulsation not resolved from certain products (mentioned L11).
- L12: What do you mean by "active phase" here? This terminology is specific to active/quiescent phases of surges. I recommend using "velocity pulsation" given the framing in the introduction.
- L15: The results you present throughout the rest of the abstract suggest that the dynamics are quite active. I recommend adding a sentence prior to this with some results related to the transition to stable dynamics.

Introduction:
- L29, L34-35: State specific years instead of "recent years". 2011 is not that recent and "recent" will be even less applicable if this paper is read years from now.
- L39: Change "to hinder" to "that hinder".
- L88 and 93: State specific years rather than "present-day".

Methods:
- L99-100: Change "surface ice velocities" to "ice surface velocities" and state here what techniques are used to derive them. Feature-tracking? InSAR?
- L103: How many 30 m DEMs in the set?
- L106-107: How much lower is the error compared to NASADEM? Were the reference measurements made *in situ*? In what time frame?
- L195: Recommend changing to "Ice surface velocity" or just "Surface velocity" here and elsewhere throughout the paper. The latter is consistent with the Section 3.2 heading in the results.
- L207: Please clarify units for the 11 x 11 window (pixels or meters).
- L223: Please justify the time separations of 5-100 and 300-430 days. Why were 100-300 day separations removed?
- L228: What was the typical variance in velocity values across the four bands? It would be convincing to report the value here.
- L292-295: Clarify what you mean by "consistency" of remote sensing data here. From looking through your Appendix D, it seems as if you are assessing both the accuracy (by confirming that velocity ratios are within physically reasonable values) and the temporal

variation in velocity / volume changes. Should these values be temporally consistent considering the dynamic pulsation just prior to those years (2000-2003)?

Results:
- L299: A length change uncertainty of 4 m (0.03%) seems unrealistically small. User error uncertainty in manual delineations is typically at least one pixel (Paul et al., 2013). For the all images other than the 0.5 m resolution images, the uncertainty should be the GSD at minimum.
- L303: Was the wave of active ice observed through velocity datasets? If so, I recommend moving this down to that section.
- L306: Oscillation in glacier length of what magnitude? It would be helpful to list a typical range here.
- L314, 315 and elsewhere: Replace "/" with dashes for the year range.
- L314: Rephrase "a gradual (but faster) slowdown" to "a faster—yet gradual—slowdown is visible".
- L342: Rephrase "abovementioned pulsations" to "pulsations mentioned above".
- L344: Add commas before and after "albiet widespread".
- L345-346: You have not yet defined what a "reservoir region" is to your readers. Similarly, you have not defined active versus quiescent phases. Please define these terms to readers in the introduction.

Discussion:
- L372: Ice redistribution due to the pulsation is a relationship of note between ice thickness and velocity. Perhaps you mean "no positively-correlated relationship" rather than "no direct relationship"
- L384-385: Would like to see the 2022-23 velocities plotted alongside the older SPOT- and IRS-derived velocities where they overlap on the glacier to better show the attainment of a new velocity peak.
- L426-427: List the value, error range, and uncertainty in this sentence.

Conclusions:
- L550: Where will the DEMs and ortho images be made available? A data repository?

**Figures and Tables:**
- Table 1: Recommend converting all spatial resolutions into meters for ease of comparison.
- Table 2: For the mono scene GSDs, replace the slashed entries (e.g., 5/10) with "and" or "or" (e.g., 5 or 10) for clarity.
- On all figures: Recommend adding bounding bars or a patch over regions of time series plots that correspond to the observed pulsations.
- Figure 2: The cyclic colormap makes the most recent (e.g., 2023) traces and the oldest traces (e.g., 1968) difficult to distinguish. Please change to a sequential colormap, keeping in mind what color schemes are colorblind friendly.
- Figure 5: Your panel ordering (vertical, as two columns) is different from in Figure 4 (horizontal, as two rows). Keep them consistent.
- Figure 6: Consider moving this figure to the Appendix or supplement since it is not critical to the main story.

**Appendices:**

- Appendix D, L636: Are there existing data to compare with to comment on whether these ice influx values are reasonable?
- Appendix E: The text in this appendix section would be appropriate to include in the main text in Section 4.4. The figures could remain in the Appendix for readability purposes.

**References:**

Paul F and 19 others (2013) On the accuracy of glacier outlines derived from remote-sensing data. Annals of Glaciology 54(63), 171–182.

---

## Referee Comment (RC4)

**Overall comments**

This paper effectively utilises a wide variety of remote sensing datasets and methods to enhance our understanding of the dynamics of Abramov glacier and address gaps in the existing observational record. The 55-year compilation of changes in glacier velocity, elevation, and terminus position convincingly demonstrates that this glacier undergoes cyclical dynamic instabilities, despite its use as a reference glacier for mass balance in this region. The manuscript is well-written, with the authors thoroughly explaining their methodology and the quantification of uncertainties. These methods have the potential for broader application, and the high level of detail achieved could be used to update regional inventories of surge-type glaciers, which likely overlook the dynamic instabilities of several smaller glaciers. Furthermore, this paper raises the question of whether other reference glaciers experience unstable flow, a possibility that a wider application of this approach might reveal. I recommend some minor revisions as outlined in the specific comments below.

**Specific comments**

Introduction:

- L41–65: In this paragraph, it would be beneficial to include more general information about glacier surging in Central Asia, such as the known ranges for the lengths of the active phase, quiescent phase, and recurrence intervals of surge-type glaciers in this region.
- L75–77: Glacier pulses have also been described in the Canadian Arctic in this paper (Van Wychen et al., 2016; https://doi.org/10.1002/2015JF003708). It may be worth citing this paper.
- L81: Consider slightly expanding on "a sudden shift in basal condition" for clarity.

Methods:

- L99–100: Refer to Section 2.3.2 here, as this statement mentions a newly produced dataset rather than analysis-ready data: "We used these scenes to derive surface ice velocities at annual frequency (Sect. 2.3.2)."
- L188: Change "aggregation polygons" to "aggregated polygons"
- L195: Change the title of section 2.3.2 to "Ice surface velocity" or "Glacier surface velocity". Make this change throughout the text wherever "surface ice velocity" is mentioned.
- L201: Spell out "pixel", i.e., "128 pixel window size"

Results:

- L299: The median rate would be a more appropriate measure than the mean rate, as it is less sensitive to outliers and would therefore be less skewed by the two periods of terminus advance. I therefore recommend using the median instead of the mean.
- L300: For added clarity, change "front advance" to "terminus advance" or "glacier front advance".
- L313: Once again, the median would be a more appropriate measure than the mean represent these velocities.
- L314–315: Use "–" for the date ranges rather than "/" here and throughout the text (e.g., "1996–1997").

- L343–344: "…ice thickness significantly increased at the terminus, reaching 90 ± 5 m in the first phase and 39 ± 4 m in the second one": it is unclear whether you are reporting ice thickness values here or changes in ice thickness (dh). If you are reporting the latter, these maximum values of thickness change may be due to the glacier advancing over previously unglaciated terrain, which should be mentioned in the text if this is the case. For better representation of the overall trends of glacier thickness changes, you should also report median thickness changes over the terminus region and at higher elevations during both the active and quiescent phases, rather than just mentioning the maximum values.
- L349: Change "the lowest 0.6 km$^2$ before front advance" to "the lowest 0.6 km$^2$ before the glacier front advance".

Discussion:

- L381–382: Consider citing this paper by Hoinkes (1969), which also supports the claim that negative mass balance conditions can transition a glacier away from unstable flow conditions: https://doi.org/10.1139/e69-086.
- L386: I suggest also citing Copland et al. (2011) here, as they attributed increased surging activity in the Karakoram during the 1990s and early 2000s to increased precipitation: https://doi.org/10.1657/1938-4246-43.4.503.
- L387: "…which was quantified at about 50 % since the 1970s in the upper accumulation area (Kronenberg et al., 2021)": mention the specific time interval over which this increase in net annual accumulation rates was quantified.
- L442–443: To slightly enhance clarity, I suggest changing the sentence as follows: "For that period, the temperature-index approach in Barandun et al. (2018) — which is constrained by transient snow line elevation from Landsat — provides the most comparable results."
- If you have space for it in the final manuscript, consider including a short section in the discussion that compares Abramov Glacier to the behavior of other surge-type glaciers in the region. This comparison could provide valuable context for understanding the unique dynamics of Abramov glacier, namely in relation to the frequency and magnitude of surges, active and quiescent phase durations, and responses to climate variability. Highlighting similarities and differences with other glaciers can also help elucidate the underlying mechanisms driving glacier behavior in this specific geographic and climatic setting.

**Figures and tables**

- Figure 1: Specify the source of the glacier outline used in this figure. Is it from RGI 7.0, or what is it manually created for this study?
- Table 1: To enhance clarity and make it easier for readers to compare the data across different platforms and sensors, I suggest providing consistent units of measurement for resolution (i.e., use either meters (m) or arc-seconds or (")).
- Figure 5: The labels should be corrected from "(c) Active phase of 2000–2003. (d) Quiescence over 2003–2020" should be "(e) Active phase of 2000–2003. (f) Quiescence over 2003–2020" to match the letters in the figure.
- Figure 5: Change "by a same amount" to "by the same amount" in the last sentence of the figure caption.
- Table 3: Rows are not aligned. Ensure this is fixed in the final version of the paper.

**References**

Copland, L., Sharp, M. J., & Dowdeswell, J. A. (2003). The distribution and flow characteristics of surge-type glaciers in the Canadian High Arctic. *Annals of Glaciology*, *36*, 73-81. https://doi.org/10.3189/172756403781816301

Hoinkes, H. C. (1969). Surges of the Vernagtferner in the Ötztal Alps since 1599. *Canadian Journal of Earth Sciences*, *6*(4), 853–861. https://doi.org/10.1139/e69-086

Van Wychen, W., Davis, J., Burgess, D. O., Copland, L., Gray, L., Sharp, M., & Mortimer, C. (2016). Characterizing interannual variability of glacier dynamics and dynamic discharge (1999–2015) for the ice masses of Ellesmere and Axel Heiberg Islands, Nunavut, Canada. *Journal of Geophysical Research: Earth Surface*, *121*(1), 39–63. https://doi.org/10.1002/2015JF003708

---

## Author Comment (AC1)

**Five decades of Abramov glacier dynamics reconstructed with multi-sensor optical remote sensing**

Enrico Mattea *et al.*

**Reply to reviewer 1**

*The authors compile a 50+ year dataset of kinematic changes of Abramov glacier, filling in gaps in the in situ observational record using a variety of remote sensing datasets. Overall, the manuscript is well-written and demonstrates how more detailed datasets of glacier kinematics can reveal novel dynamic behavior that may complicate mass balance studies. I applaud the authors for the thoroughness of their data processing and presentation of the methodology. I recommend a minor revision of the manuscript with the specific comments listed point by point below.*

We would like to thank the reviewer for the positive and constructive review of our manuscript. Below, we provide point-by-point answers to the comments. Any comments which are not mentioned here are considered accepted and fully implemented in the revised manuscript. The review text is reported in *black italic,* while our responses are in blue.

***Specific comments:***
*Abstract:*
- *L9: It would be helpful to mention which "archives" are used in this study, especially since your results show a newly-discovered pulsation not resolved from certain products (mentioned L11).*

We agree with the reviewer on this point. However, we think that the full list of archives (with more than a dozen different sources) would be too long for an abstract. In the revised manuscript we mention examples of the most significant archives – Key Hole (KH), SPOT and RapidEye.

- *L15: The results you present throughout the rest of the abstract suggest that the dynamics are quite active. I recommend adding a sentence prior to this with some results related to the transition to stable dynamics.*

In the revised manuscript, we are rephrasing as "However, we also find a decreasing magnitude and increasing duration of the pulsations, suggestive of a potential ongoing transition towards more stable dynamics."

*Introduction:*
- *L29, L34-35: State specific years instead of "recent years". 2011 is not that recent and "recent" will be even less applicable if this paper is read years from now.*

We agree with the reviewer on this point. However, at L34-35 (the list of recent remote sensing studies over Central Asia), the years of the cited studies are already provided in the citations themselves, therefore we are dropping the mention "In recent years" but we are opting to not repeat the year specification.

- *L88 and 93: State specific years rather than "present-day".*

We are dropping the sentence at L88 entirely, since the study actually includes analyses of all

sub-periods. At L93, we are rewording as "the build-up to a third one during the 2010s and early 2020s".

*Methods:*
- *L99-100: Change "surface ice velocities" to "ice surface velocities" and state here what techniques are used to derive them. Feature-tracking? InSAR?*

In the revised manuscript we are adding this information (frequency-domain correlation).

- *L103: How many 30 m DEMs in the set?*

In the revised manuscript we are adding this information (11 DEMs, although – as stated in the subsequent line – only 4 were usable).

- *L106-107: How much lower is the error compared to NASADEM? Were the reference measurements made in situ? In what time frame?*

The studies we cite in this section (Fahrland, 2022; Li et al., 2022; Okolie et al., 2024) report a large number of numeric values for the comparison of the Copernicus DEM to NASADEM – including various terrain types and reference data. Here, we just convey that the overall precision and accuracy of the Copernicus DEM are in most cases considered superior to NASADEM; in the subsequent sentences of our manuscript we explain why this is not the case in the Abramov glacier region. Thus, we believe that providing detailed numbers or additional information from the papers comparing these two DEMs goes beyond what is relevant in this section.

- *L207: Please clarify units for the 11 x 11 window (pixels or meters).*

In the revised manuscript we are adding this information (pixels).

- *L223: Please justify the time separations of 5-100 and 300-430 days. Why were 100-300 day separations removed?*

In the revised manuscript we are adding this information (in order to minimize the variability of surface characteristics and solar illumination).

- *L228: What was the typical variance in velocity values across the four bands? It would be convincing to report the value here.*

In the revised manuscript we are adding this information (1.1 m yr$^{-1}$, computed as standard deviation of velocity within the four velocity rasters of one year, averaged over all cells within the Abramov glacier outline).

- *L292-295: Clarify what you mean by "consistency" of remote sensing data here. From looking through your Appendix D, it seems as if you are assessing both the accuracy (by confirming that velocity ratios are within physically reasonable values) and the temporal*

*variation in velocity / volume changes. Should these values be temporally consistent considering the dynamic pulsation just prior to those years (2000-2003)?*

In Appendix D we are checking consistency of our results only via the ratio of depth-averaged to surface velocity – this ratio is computed from the estimated velocity and thickness changes. The verification is that such a ratio falls within physically reasonable values, in particular, as expected, we obtain a high ratio for the late stage of the pulsation (2003 to 2004) and a lower ratio for the two subsequent one-year periods. As such, the occurrence of the pulsation until 2004 offers the possibility to check our results under a wide range of values of the ice flux (see Appendix D). In the revised manuscript we are clarifying this sentence and the first paragraph of Appendix D.

- *L299: A length change uncertainty of 4 m (0.03%) seems unrealistically small. User error uncertainty in manual delineations is typically at least one pixel (Paul et al., 2013). For the all images other than the 0.5 m resolution images, the uncertainty should be the GSD at minimum.*

We agree with the reviewer that in general the uncertainty in manual delineations should be at least one pixel. Indeed, the length change under question (-1106 ± 4 m over the period 1968 to 2023) was computed from a CORONA image at 1.8 m and a Pléiades image at 0.5 m (Table 2): the uncertainty of 4 m is more than double the value of both GSDs. We also note that the glacier length change is computed with the rectilinear box method (Sect. 2.3.1), which performs aggregation of the changes (and thus their uncertainties) over the full width of the glacier terminus: thus, in some cases it could be possible to achieve a smaller uncertainty of length change than the pixel size of either image in the pair.

- *L303: Was the wave of active ice observed through velocity datasets? If so, I recommend moving this down to that section.*

The wave is indeed visible in velocity datasets (Fig. 4a). In the revised manuscript, we are moving this description to the section on velocity results.

- *L306: Oscillation in glacier length of what magnitude? It would be helpful to list a typical range here.*

In the revised manuscript we are adding this information (30 to 50 m).

- *L345-346: You have not yet defined what a "reservoir region" is to your readers. Similarly, you have not defined active versus quiescent phases. Please define these terms to readers in the introduction.*

In the revised manuscript we are defining "reservoir region" in the Introduction. We note that "active phase" and "quiescence" are already introduced, at L44-45. We are expanding their definition.

*Discussion:*
- *L372: Ice redistribution due to the pulsation is a relationship of note between ice thickness and velocity. Perhaps you mean "no positively-correlated relationship" rather than "no*

*direct relationship"*

In the revised manuscript we are updating the text as suggested.

- *L384-385: Would like to see the 2022-23 velocities plotted alongside the older SPOT- and IRS-derived velocities where they overlap on the glacier to better show the attainment of a new velocity peak.*

Unfortunately, the 2022-23 velocity peak (Fig. 4c) is occurring in a region where SPOT and IRS velocities are missing due to sensor saturation and lower snowlines in the early 2000s (L329).

- *L426-427: List the value, error range, and uncertainty in this sentence.*

An exact calculation of the error range and uncertainty is unfortunately not possible in our case, because there is no information on the distribution and accuracy of the measurements of ice thickness used in the calculations by Emelyianov *et al.* (1974, from radio-echo surveys performed in the 1960s) and for the bed DEM used in our study (from radar surveys of 1986). Ice thickness measured by radar, later interpolated into a map and subsequently converted into a DEM from the contours of such a map has several poorly-constrained sources of uncertainty: among them performance of the early radar systems, used wavelength, manual picking of reflectors, horizontal distance from the measured point, re-interpolation between contour lines. Thus, here we simply did a rough estimation of overall uncertainties from a literature-based value of 20 % for the ice thickness uncertainties (L636; Grab *et al.*, 2021). We also note that the statement by Emelyianov *et al.* (1974) about the evolution of ice volume during the first pulsation is provided without any uncertainty or absolute values; in particular, the uncertainty in the calculation of total ice volume by the authors is not known but probably quite high, since the interpolation from measured points was likely performed manually. The authors simply report "a doubling of the total ice volume over the first 8 months": L426). By introducing the estimated 20 % uncertainty in all volume calculations, we obtain a range of 70 to 160 % volume increase (over January-August 1973) in the results of Emelyianov *et al.* (1974), and of 40 to 80 % volume increase from the remote sensing data.
In the revised manuscript, we are including a summary of these considerations to explain our reasoning and the uncertainty estimates.

*Conclusions:*
- *L550: Where will the DEMs and ortho images be made available? A data repository?*

The DEMs and ortho images are already available to reviewers through the review platform. Upon publication, they will be made available via Zenodo.

*Figures and Tables:*
- *Table 1: Recommend converting all spatial resolutions into meters for ease of comparison*

In this table, we are reporting the original resolution of the datasets as they are provided. The global NASADEM and Copernicus DEMs are provided in equirectangular projection (EPSG:4326) for which the actual resolution can only be expressed in arc-seconds (resolution in meters is not spatially constant), while all other products use projected coordinate systems whose resolution can only be expressed in meters. As an alternative, in the revised manuscript we are adding to the Table caption information about the metric resolution which is commonly used at the mid-latitudes when

re-projecting global DEMs to projected coordinate systems (30 m for 1", 90 m for 3").

- *Figure 2: The cyclic colormap makes the most recent (e.g., 2023) traces and the oldest traces (e.g., 1968) difficult to distinguish. Please change to a sequential colormap, keeping in mind what color schemes are colorblind friendly.*

We agree with the reviewer that cyclic colormaps can sometimes make visualization difficult. However, in this case the large number of colors would make a sequential color map even harder to read than a cyclic one. Moreover, in the figure, lines with similar colors (the most recent and oldest traces) are also separated in space by more than 1 km, and the annotated legend on the left of the figure also describes the direction of the changes, making it clear which line corresponds to which year. We tested several alternative color schemes for the figure, but could not find a more satisfactory solution for visualizing all the digitized outlines.

*Appendices:*
- *Appendix D, L636: Are there existing data to compare with to comment on whether these ice influx values are reasonable?*

There are some Soviet-era estimates of ice discharge at Abramov glacier in the years surrounding the 1970s pulsation, but they refer to flux gates located several km upstream of the region where we have remote sensing data for our estimates. Thus, even though the order of magnitude is the same as our results (between 1 and 10 million $m^3$ per year), we see limited value in reporting such a comparison.

**References**

Emelyianov, Y., Nozdriukhin, V., and Suslov, V.: Dynamics of the Abramov Glacier during the 1972-1973 surge, Materialy gliatsiologicheskikh issledovaniy, 24, 87–96, 1974.

Fahrland, E.: Copernicus Digital Elevation Model Product Handbook (v4.0), Tech. Rep. AO/1-9422/18/I-LG, Airbus Defence and Space GmbH, Taufkirchen, Germany, https://spacedata.copernicus.eu/documents/20123/121239/GEO1988-CopernicusDEM-SPE-002_ProductHandbook_I4.0.pdf, 2022.

Grab, M., Mattea, E., Bauder, A., Huss, M., Rabenstein, L., Hodel, E., Linsbauer, A., Langhammer, L., Schmid, L., Church, G., Hellmann, S., Délèze, K., Schaer, P., Lathion, P., Farinotti, D., and Maurer, H.: Ice thickness distribution of all Swiss glaciers based on extended ground-penetrating radar data and glaciological modeling, Journal of Glaciology, 67, 1074–1092, https://doi.org/10.1017/jog.2021.55, 2021.

Li, H., Zhao, J., Yan, B., Yue, L., and Wang, L.: Global DEMs vary from one to another: an evaluation of newly released Copernicus, NASA and AW3D30 DEM on selected terrains of China using ICESat-2 altimetry data, International Journal of Digital Earth, 15, 1149–1168, https://doi.org/10.1080/17538947.2022.2094002, 2022.

Okolie, C. J., Mills, J. P., Adeleke, A. K., Smit, J. L., Peppa, M. V., Altunel, A. O., and Arungwa, I. D.: Assessment of the global Copernicus, NASADEM, ASTER and AW3D digital elevation models in Central and Southern Africa, Geo-spatial Information Science, pp. 1–29,

https://doi.org/10.1080/10095020.2023.2296010, 2024.

---

## Author Comment (AC2)

**Five decades of Abramov glacier dynamics reconstructed with multi-sensor optical remote sensing**

Enrico Mattea *et al.*

**Reply to reviewer 2**

*In this paper, the authors have utilized an impressive array of remote sensing datasets and applied a range of techniques to produce a 50+ year record of surface ice velocity, elevation change, and terminus position change for Abramov glacier. I found this paper very easy to read and follow, with both the data processing and characterization of uncertainty well-explained. The results are well-demonstrated, and provide good support for (almost) all of the conclusions. As such, I have only relatively minor comments on the manuscript that should be easy to address.*

We would like to thank the reviewer for the positive and constructive review of our manuscript. Below, we provide point-by-point answers to the comments. Any comments which are not mentioned here are considered accepted and fully implemented in the revised manuscript. The review text is reported in *black italic,* while our responses are in blue.

- *l. 11: "unobserved" pulsation. In the comparison with Mandychev et al. (2017), you show that those authors observed an advance of the glacier, reported as beginning in 2000 (rather than 2002, as you have shown). This seems to be a contradiction with the claim here (and in the conclusions), that this pulsation is "unobserved". The claim that this is better captured by your data/observations than in previous global datasets or other studies is not quite the same thing, so I feel that this claim should be softened somewhat.*

We agree with the reviewer on this point. In the revised manuscript, we are rephrasing the sentence as follows: "We describe at subseasonal scale a second pulsation over 2000–2005, not observed *in situ* and poorly resolved by Landsat and ASTER products"

- *l. 38-39: suggest "... found that data inconsitencies and regional simplifications hinder interpretation ..."*

In the revised manuscript, the entire sentence is being slightly reworded following comments by all reviewers.

- *l. 258: "within bins": what size are the bins used here?*

We used a constant N = 500 bins for the along-/across-track corrections. As such, the actual dimension of each bin in m depends on the along-/across-track angles of the correction, and the number of samples in each bin additionally depends on the amount of missing data within each grid. During preliminary analysis, we found very little sensitivity of our results to the size of these bins, as long as they were small enough to resolve the targeted biases and large enough to hold enough samples. In the revised manuscript, we provide this information.

- *Fig. 4: would it be possible to include different symbols/patterns to help differentiate the*

*colors here?*

In the revised manuscript, we are adding an alternating pattern of solid line / dashed line to improve differentiation.

- *Fig. 5: same comment for panels (c, d) and (g, h) as for Fig. 4*

We are not sure about the possibility to further differentiate the data points here. The plots already use a colorblind-friendly color scale and the plotted dots are geometrically separate (no overlap in panels c and d, unambiguous overlap of only SPOT and ASTER data in panels g and h, since NASADEM corresponds to a single data point). Moreover, in order to be geometrically separate, the plotted dots are not large enough to benefit from the use of multiple symbols beyond simple circles. For panels c and d, we note that the exact acquisition date of all declassified scenes is reported in Table A2.

---

## Author Comment (AC3)

**Five decades of Abramov glacier dynamics reconstructed with multi-sensor optical remote sensing**
Enrico Mattea *et al.*

**Reply to reviewer 3**

*This paper effectively utilises a wide variety of remote sensing datasets and methods to enhance our understanding of the dynamics of Abramov glacier and address gaps in the existing observational record. The 55-year compilation of changes in glacier velocity, elevation, and terminus position convincingly demonstrates that this glacier undergoes cyclical dynamic instabilities, despite its use as a reference glacier for mass balance in this region. The manuscript is well-written, with the authors thoroughly explaining their methodology and the quantification of uncertainties. These methods have the potential for broader application, and the high level of detail achieved could be used to update regional inventories of surge-type glaciers, which likely overlook the dynamic instabilities of several smaller glaciers. Furthermore, this paper raises the question of whether other reference glaciers experience unstable flow, a possibility that a wider application of this approach might reveal. I recommend some minor revisions as outlined in the specific comments below.*

We would like to thank the reviewer for the positive and constructive review of our manuscript. Below, we provide point-by-point answers to the comments. Any comments which are not mentioned here are considered accepted and fully implemented in the revised manuscript. The review text is reported in *black italic,* while our responses are in blue.

*Specific comments*
*Introduction:*
- *L41–65: In this paragraph, it would be beneficial to include more general information about glacier surging in Central Asia, such as the known ranges for the lengths of the active phase, quiescent phase, and recurrence intervals of surge-type glaciers in this region.*

In the revised manuscript we are adding this information: "The inventory of surge-type glaciers in High Mountain Asia compiled by Guillet *et al.* (2022) reports active phases lasting 1 to 18 years (median = 2 years, N = 30) in Tien Shan and 1 to 19 years (median = 3 years, N = 73) in the Pamirs. Information on recurrence intervals of surging activity is scarce in that and other inventories, because usually a single surge cycle takes place during the considered time interval. Reported values range from one to five decades (Murodov *et al.*, 2024; Mukherjee *et al.*, 2017)".

- *L81: Consider slightly expanding on "a sudden shift in basal condition" for clarity.*

In the revised manuscript we are rephrasing this section to provide more information: "Using a minimal flow-line model, Glazirin *et al.* (1987) investigated the pulsation with various formulations of the basal sliding law. The best agreement with observed ice velocities was found by introducing a switch in the friction coefficient between two different values, as controlled by a threshold of basal shear stress: as such, the authors attributed the pulsation to a sudden shift in basal conditions, but the mechanism of such a shift was not examined."

*Methods:*
- *L188: Change "aggregation polygons" to "aggregated polygons"*

We are opting to use the terminology "regions of interest" instead, to maintain consistency within the rest of the text (e.g., Sect. 2.3.2).

**Results:**
- *L299: The median rate would be a more appropriate measure than the mean rate, as it is less sensitive to outliers and would therefore be less skewed by the two periods of terminus advance. I therefore recommend using the median instead of the mean.*

The mean rate reported here is not computed as the arithmetic average of each year's change, since the interval corresponding to each change is not constant (L297-298). Rather, the mean rate is derived from the total change of −1106 ± 4 m divided by 55.064 years – thus, the periods of terminus advance are not affecting the calculation. In the revised manuscript we are clarifying this by rephrasing the expression to "with a total change of −1106 ± 4 m corresponding to a mean rate of 20.09 ± 0.07 m yr$^{-1}$".

- *L313: Once again, the median would be a more appropriate measure than the mean represent these velocities.*

We are not sure that we fully understand the reviewer's comment. The velocities mentioned at L313 are not statistical aggregations of multiple measurements, but rather single measurements, computed by rescaling to 365.25 days the total displacements observed over different durations. As such, they are necessarily measurements of the mean velocity of the ice over each duration, and no median can be computed. In the revised manuscript, we are clarifying this point by using and explaining the expression "mean annual velocity" already in the methodological section 2.3.2. We note that where relevant (in the statistical aggregation of Sentinel-2 velocities) we indeed use the median (L227).

**Discussion:**
- *L343–344: "...ice thickness significantly increased at the terminus, reaching 90 ± 5 m in the first phase and 39 ± 4 m in the second one": it is unclear whether you are reporting ice thickness values here or changes in ice thickness (dh). If you are reporting the latter, these maximum values of thickness change may be due to the glacier advancing over previously unglaciated terrain, which should be mentioned in the text if this is the case. For better representation of the overall trends of glacier thickness changes, you should also report median thickness changes over the terminus region and at higher elevations during both the active and quiescent phases, rather than just mentioning the maximum values.*

In the revised manuscript we are clarifying this point: it was the increase of ice thickness (not its absolute value) which reached maximum values of 90 and 39 m, and in both cases such a maximum increase took place at locations which were glacierized both at the start and end of the period covered by the DEM differences.
Concerning the better representation of overall trends of thickness changes, we note that the median change is not a suitable estimator in our case, due to the presence of gaps in the grids of DEM difference; the spatial distribution of these gaps is not uniform, and as such, the simple median is a biased estimate of overall change. At L350-351, we are instead providing values of mean change derived by the hypsometric method (L266), which provides an unbiased estimate of the change (McNabb *et al.*, 2019). For comparability, we compute such a change over a terminus region defined to match the regions of interest used in previous studies. In the revised manuscript, we are

additionally providing the hypsometric mean change over the regions of thickening and thinning during both active and quiescence phases (the time intervals presented in Fig. 5): +29 ± 4 m and -10 ± 4 m (1972-1973), +1.5 ± 1.2 m and -24.2 ± 1.1 m (1980-2000), +9.1 ± 0.6 m and -2.8 ± 0.5 m (2000-2003), +2.0 ± 0.4 m and -19.6 ± 0.2 m (2003-2020).

- *L387: "...which was quantified at about 50 % since the 1970s in the upper accumulation area (Kronenberg et al., 2021)": mention the specific time interval over which this increase in net annual accumulation rates was quantified.*

In the revised manuscript we are adding this information (between 1970–97 and 2011–18).

- *If you have space for it in the final manuscript, consider including a short section in the discussion that compares Abramov Glacier to the behavior of other surge-type glaciers in the region. This comparison could provide valuable context for understanding the unique dynamics of Abramov glacier, namely in relation to the frequency and magnitude of surges, active and quiescent phase durations, and responses to climate variability. Highlighting similarities and differences with other glaciers can also help elucidate the underlying mechanisms driving glacier behavior in this specific geographic and climatic setting.*

We agree with the reviewer that such a comparison is valuable to better understand the mechanisms driving unstable glacier behavior in the region. We are currently working on a detailed, regional-scale investigation of unstable ice flow in the whole of Pamir-Alay, which will be the topic of an upcoming publication including a discussion of the possible mechanisms of unstable ice flow. We believe that it is a more appropriate site to discuss the similarities and differences of unstable ice flow between this specific setting and other surge-type glaciers in Central Asia.

**Figures and tables**
- *Figure 1: Specify the source of the glacier outline used in this figure. Is it from RGI 7.0, or what is it manually created for this study?*

In the revised manuscript we are providing this information – the outline was created manually (L185) from the Pléiades orthoimage of 5 September 2022.

- *Table 1: To enhance clarity and make it easier for readers to compare the data across different platforms and sensors, I suggest providing consistent units of measurement for resolution (i.e., use either meters (m) or arc-seconds or (")).*

In this table, we are reporting the original resolution of the datasets as they are provided. The global NASADEM and Copernicus DEMs are provided in equirectangular projection (EPSG:4326) for which the actual resolution can only be expressed in arc-seconds, while all other products use projected coordinate systems whose resolution can only be expressed in meters. As an alternative, in the revised manuscript we are adding to the Table caption some information about the resolution which is commonly used in the mid-latitudes when re-projecting global DEMs to projected coordinate systems (30 m for 1", 90 m for 3"; we note that this is an approximation as the proper conversion is not spatially uniform).

- *Figure 5: The labels should be corrected from "(c) Active phase of 2000–2003. (d) Quiescence over 2003–2020" should be "(e) Active phase of 2000–2003. (f) Quiescence over 2003–2020" to match the letters in the figure.*

We thank the reviewer for catching this, it is being corrected in the revised manuscript.

- *Figure 5: Change "by a same amount" to "by the same amount" in the last sentence of the figure caption.*

We believe that an indefinite article is more appropriate here since the amount under question is unknown.

- *Table 3: Rows are not aligned. Ensure this is fixed in the final version of the paper.*

We are not sure that we fully understand the reviewer's comment. The table presents geodetic mass balance computed over several intervals, and the first column gives the boundaries of such intervals. As such, we find it reasonable to align the rows of the other columns (which provide values referring to each interval) to the middle of the intervals of the first column. We note that the first column has one more row compared to all others (7 dates, which define 6 consecutive time intervals).

**References**

Glazirin', G., Kamnyansky, G., Mazo, A., Nozdriukhin, V., and Salamatin, A.: Mechanism of the Abramov glacier advance in 1972-1975, Materialy gliatsiologicheskikh issledovaniy, 60, 84–90, 1987.

Guillet, G., King, O., Lv, M., Ghuffar, S., Benn, D., Quincey, D., and Bolch, T.: A regionally resolved inventory of High Mountain Asia surge-type glaciers, derived from a multi-factor remote sensing approach, The Cryosphere, 16, 603–623, https://doi.org/10.5194/tc-16-603-2022, 2022.

McNabb, R., Nuth, C., Kääb, A., and Girod, L.: Sensitivity of glacier volume change estimation to DEM void interpolation, The Cryosphere, 13, 895–910, https://doi.org/10.5194/tc-13-895-2019, 2019.

Mukherjee, K., Bolch, T., Goerlich, F., Kutuzov, S., Osmonov, A., Pieczonka, T., and Shesterova, I.: Surge-Type Glaciers in the Tien Shan (Central Asia), Arctic, Antarctic, and Alpine Research, 49, 147–171, https://doi.org/10.1657/AAAR0016-021, 2017.

Murodov, M.; Li, L.; Safarov, M.; Lv, M.; Murodov, A.; Gulakhmadov, A.; Khusrav, K.; Qiu, Y. A Comprehensive Examination of the Medvezhiy Glacier's Surges in West Pamir (1968–2023). *Remote Sens., 16(10),* 1730, 2024. https://doi.org/10.3390/rs16101730